# Dephosphocholination by Legionella effector Lem3 functions through remodelling of the switch II region of Rab1b

Marietta S. Kaspers [1,6], Vivian Pogenberg [1,6], Christian Pett [2], Stefan Ernst[3], Felix Ecker [4], Philipp Ochtrop [2], Michael Groll [4], Christian Hedberg [2] & Aymelt Itzen [1,5] ✉

Bacterial pathogens often make use of post-translational modifications to manipulate host cells. *Legionella pneumophila*, the causative agent of Legionnaires disease, secretes the enzyme AnkX that uses cytidine diphosphate-choline to post-translationally modify the human small G-Protein Rab1 with a phosphocholine moiety at Ser76. Later in the infection, the *Legionella* enzyme Lem3 acts as a dephosphocholinase, hydrolytically removing the phosphocholine. While the molecular mechanism for Rab1 phosphocholination by AnkX has recently been resolved, structural insights into the activity of Lem3 remained elusive. Here, we stabilise the transient Lem3:Rab1b complex by substrate mediated covalent capture. Through crystal structures of Lem3 in the apo form and in complex with Rab1b, we reveal Lem3's catalytic mechanism, showing that it acts on Rab1 by locally unfolding it. Since Lem3 shares high structural similarity with metal-dependent protein phosphatases, our Lem3:Rab1b complex structure also sheds light on how these phosphatases recognise protein substrates.

*Legionella pneumophila* is a gram-negative bacterium that causes Legionnaires' disease by infecting human alveolar macrophages. After phagocytosis by the immune cell, the bacterium escapes the cellular defence mechanisms by forming a replicative organelle referred to as the Legionella containing vacuole (LCV). Crucial to the evasion of host defence is the release of about 330 bacterial proteins (also known as bacterial effectors) through the type IV secretion system[1–4].

Among the many host targets of the bacterial effectors are Rab proteins, in particular the small G-protein Rab1b, which acts as a central regulatory hub in vesicular trafficking[5]. Rab1b functions as a molecular switch that is in the inactive state when binding to guanosine diphosphate (GDP) and active when binding to guanosine triphosphate (GTP). In the active state, Rab1b promotes vesicular trafficking from the endoplasmic reticulum (ER) to the Golgi apparatus through the recruitment of GTP-state specific interaction partners. Rab1b activation is catalysed by guanine nucleotide exchange factors (GEFs) that replaces the tightly bound GDP with GTP, whereas inactivation is stimulated by GTPase activating proteins (GAPs) that produce the inactive form by accelerating the intrinsic GTP-hydrolysis activity of Rab1b[6,7]. Active Rab-proteins (and thus also Rab1b) are localized to intracellular membranes by means of post-translationally attached geranylgeranyl moieties[8]. In the inactive state, Rab1b is extracted from the membrane to the cytosol through complexation with the protein GDP dissociation inhibitor (GDI).

[1]Institute of Biochemistry and Signal Transduction, University Medical Centre Hamburg-Eppendorf (UKE), Martinistr. 52, 20246 Hamburg, Germany. [2]Chemical Biology Center (KBC), Department of Chemistry, Umeå University, Linnaeus väg 10, 90187 Umeå, Sweden. [3]Center for Integrated Protein Science Munich (CIPSM), Department Chemistry, Technical University of Munich, Lichtenbergstrasse 4, 85747 Garching, Germany. [4]Center for Protein Assemblies, Technical University of Munich, Ernst-Otto-Fischer-Str. 8, 85748 Garching, Germany. [5]Centre for Structural Systems Biology, University Medical Centre Hamburg-Eppendorf (UKE), Martinistr. 52, 20246 Hamburg, Germany. [6]These authors contributed equally: Marietta S. Kaspers, Vivian Pogenberg. ✉e-mail: a.itzen@uke.de

During *Legionella* infection, Rab1b proteins are directed towards and activated at the LCV, leading to the rerouting of ER-derived vesicles to this compartment[9,10]. In total, six different *Legionella* effectors are involved in manipulating Rab1b in this process: DrrA/SidM contains a GEF and an AMPylation domain, leading to Rab1b activation and AMPylation, respectively[11–13]. AMPylation is a post-translational modification (PTM) in which adenosine monophosphate (AMP) is transferred from adenosine triphosphate (ATP) to proteins. Here, Y77 of Rab1b gets AMPylated, likely leading to changing interaction profiles of Rab1b and activating it[14]. Also, the effector LidA from *Legionella* can bind AMPylated and non-AMPylated Rab1b[12]. At later stages of infection, the *Legionella* effectors SidD and LepB lead to deAMPylation and GTP-hydrolysis, thereby reconstituting the inactive state of Rab1b[15,16].

In addition, Rab1b undergoes another PTM: The effector AnkX makes use of the nucleotide cytidine diphosphate (CDP)-choline and transfers a phosphocholine (PC) group to $S76_{Rab1b}$ (a process referred to as phosphocholination), resulting in a phosphodiester-linked choline modification at this site[17]. Phosphocholination impairs Rab1b interactions with GAPs and GDI[18]. In addition, the *Legionella* protein Lem3 can hydrolytically cleave the phosphocholine group (i.e. dephosphocholination) and reconstitute the unmodified Rab1b protein[18,19]. It has to be noted that protein phosphocholination and its enzymatic mechanism are barely studied in human proteins.

Our recent work has provided detailed functional and structural data on the mechanism of Rab1b phosphocholination by *Legionella* effector AnkX[20]. However, the structural mechanism by which Lem3 cleaves the phosphocholine group has not yet been analysed due to the transient nature of the complex. Protein remote homology detection and three-dimensional structure prediction of Lem3 using HHpred strongly suggested that the protein shares structural homology to the $Mg^{2+}/Mn^{2+}$-dependant protein phosphatases (PPMs)[21,22]. Phosphatases hydrolytically cleave phosphorylated amino acids. In this context, the PPM1A-family (also referred to as PP2C) is specific for phosphorylated serine (pS) and phosphorylated threonine (pT). Dephosphorylation is dependent on divalent cations (frequently $Mg^{2+}$ or $Mn^{2+}$) in the enzymes' active centre that are coordinated by a network of aspartates and glutamates[23,24]. It has been suggested that the cations stimulate catalysis through charge compensation of the phosphate and water activation as these entities are also coordinated to the metal centre[25].

The structural investigation of the complex between Rab1b and Lem3 is hampered by the low affinity of their interaction, thereby not permitting quantitative complex preparation. In this work, we applied a site-specific covalent method to link Rab1b and Lem3 through a phosphocholine derivative. Thanks to this approach, we were able to solve the crystal structures of the Lem3 apo-form and the covalent Lem3:Rab1b complex.

## Results

### The Lem3 crystal structure

In order to obtain structural insights into Lem3, we crystallised the full-length protein comprising the amino acids (aa) 1-570 ($Lem3_{FL}$) and a shortened construct (aa 21-486, $Lem3_{21-486}$). $Lem3_{21-486}$ possesses full catalytic activity in regard to dephosphocholination of Rab1b phosphocholinated at Ser76, indicating that the N- and C-terminal regions of Lem3 are not involved in catalysis (Fig. 1a). We solved the structures of $Lem3_{FL}$ and $Lem3_{21-486}$ at 3.6 Å and 2.2 Å resolution, respectively. The $Lem3_{21-486}$ structure was solved experimentally using the anomalous dispersion from heavy atoms incorporated to the protein by soaking (Supplementary Table 1). In parallel, $Lem3_{21-486}$ was modelled using AlphaFold2 (AF2)[26]. A superimposition of the $Lem3_{21-486}$ crystal and the AF2-predicted structure revealed high similarity (0.88 Å RMSD (root-mean-square deviation) on 444 superimposed Cα-atoms) (Supplementary Fig. 1a).

Furthermore, we were able to solve the structure of $Lem3_{FL}$ by molecular replacement using the short construct structure and further extend the building of the C-terminal part with help of the AF2 model of $Lem3_{FL}$ (Fig. 1b, Supplementary Table 1)[27].

Overall, Lem3 has a scalene triangular shape, formed by 17 β-strands and 20 α-helices (Fig. 1b, c). Metaphorically, the core domain folds into a fist and a raised thumb, the latter consisting of helices α9-α13. The extended α-helix α9 appears to play the role of the first metacarpal bone which links the thumb to the palm. $Lem3_{FL}$ extends the fist opposite from the thumb by three α-helices (α18-α20, aa 495-567). Together with α-helices α14-α17, these three helices form a helical bundle consisting of helices α14-α20 (Fig. 1b, c).

### The Lem3 structure reveals similarity to PPM phosphatases

A structural comparison of Lem3 using PDBeFold revealed that the closest structural homologue of Lem3 is the deAMPylase domain of the *Legionella* effector SidD (Fig. 2a, b)[28]. Other homology results demonstrate that Lem3 adopts a phosphatase fold and is highly similar to PPMs, in particular to the human PPM1A (also referred to as PP2Cα) (PDB ID: 4RA2) (RMSD for $Lem3_{21-486}$ and 4RA2: 2.24 Å on 187 superimposed Cα-atoms) (Fig. 2b, c)[29]. Human PPM1A contains a conserved β-sheet region consisting of 11 β-strands localised in the centre of the protein. The β-strands are arranged in two opposing β-sheets forming a cleft between them (referred to as the core). This arrangement is flanked by several α-helices. Two manganese ions are located at the base of the β-sheet cleft and coordinated by six amino acids, i.e. four aspartate side chains ($D38_{PPM1A}$, $D60_{PPM1A}$, $D239_{PPM1A}$, $D282_{PPM1A}$), the side chain of $E37_{PPM1A}$, and the backbone carbonyl of $G61_{PPM1A}$ (Supplementary Fig. 1b).

$Lem3_{21-486}$ differs from the conserved PPM1A fold in minor details (Fig. 2b, c): The central β-sheet core consists of two additional β-strands (β1 and β19), thereby broadening the cleft. The surrounding α-helices are more numerous and mostly located at the base and top of the cleft. Altogether, this particular arrangement represents the above-mentioned fist region of $Lem3_{21-486}$. The spatial orientation of the additional α-helices forming the thumb shape a deep cavity located at the interception of the fist and the thumb (referred to as the hollow region). The base of this hollow region, which is located closely to the β-sheet cleft in the active centre, harbours the metal ions (Fig. 2d). The position of the metal ions, referred to as M1 and M2, corresponds to the position of the manganese ions in the PPM1A crystal structure (PDB ID: 4RA2). As verified by CheckMyMetal, these metal ions are embedded in an octahedral cluster which is characteristic of the coordination of $Mg^{2+}$, $Mn^{2+}$, or $Ca^{2+}$ ions[30]. This metal cluster is surrounded by a scaffold of oxygen atoms provided by the side chains of four aspartate residues ($D82_{Lem3}$, $D105_{Lem3}$, $D254_{Lem3}$, and $D394_{Lem3}$), the backbone carbonyl of a glycine ($G106_{Lem3}$) as well as six water molecules (Fig. 2b–d, Supplementary Fig. 1b)[31]. More precisely, M1 is coordinated by $D105_{Lem3}$, $D254_{Lem3}$, $D394_{Lem3}$, and three water molecules. M2 is coordinated by $D105_{Lem3}$, the main chain-oxygen atom of $G106_{Lem3}$, and four solvent molecules. One of these water molecules is shared with M1. In addition, $D82_{Lem3}$ further stabilises one molecule of water linked to M1 and one connected to M2 via hydrogen bonds.

In addition to the β-strands β1 and β19 that broaden the core β-sheets, $Lem3_{21-486}$ has six other β-strands (β2, β5, β12, β13, β16, β17) forming two additional β-sheets (one β-sheet consists of β2-β5 and the second of β12-β13 and β16-β17) located in the fist region (Fig. 1b, c, Supplementary Fig. 1a). $PPM1A_{2-368}$ (PDB ID: 4RA2) and $Lem3_{FL}$ both have C-terminal α-helical bundles consisting of three (α10-α12) and seven (α14-α20) helices, respectively. The bundles are differently located towards the central β-sheet core but share some structural similarities (Supplementary Fig. 1c).

In summary, the Lem3 crystal structures reveal a conserved PPM-like fold with metal ions at the catalytic centre.

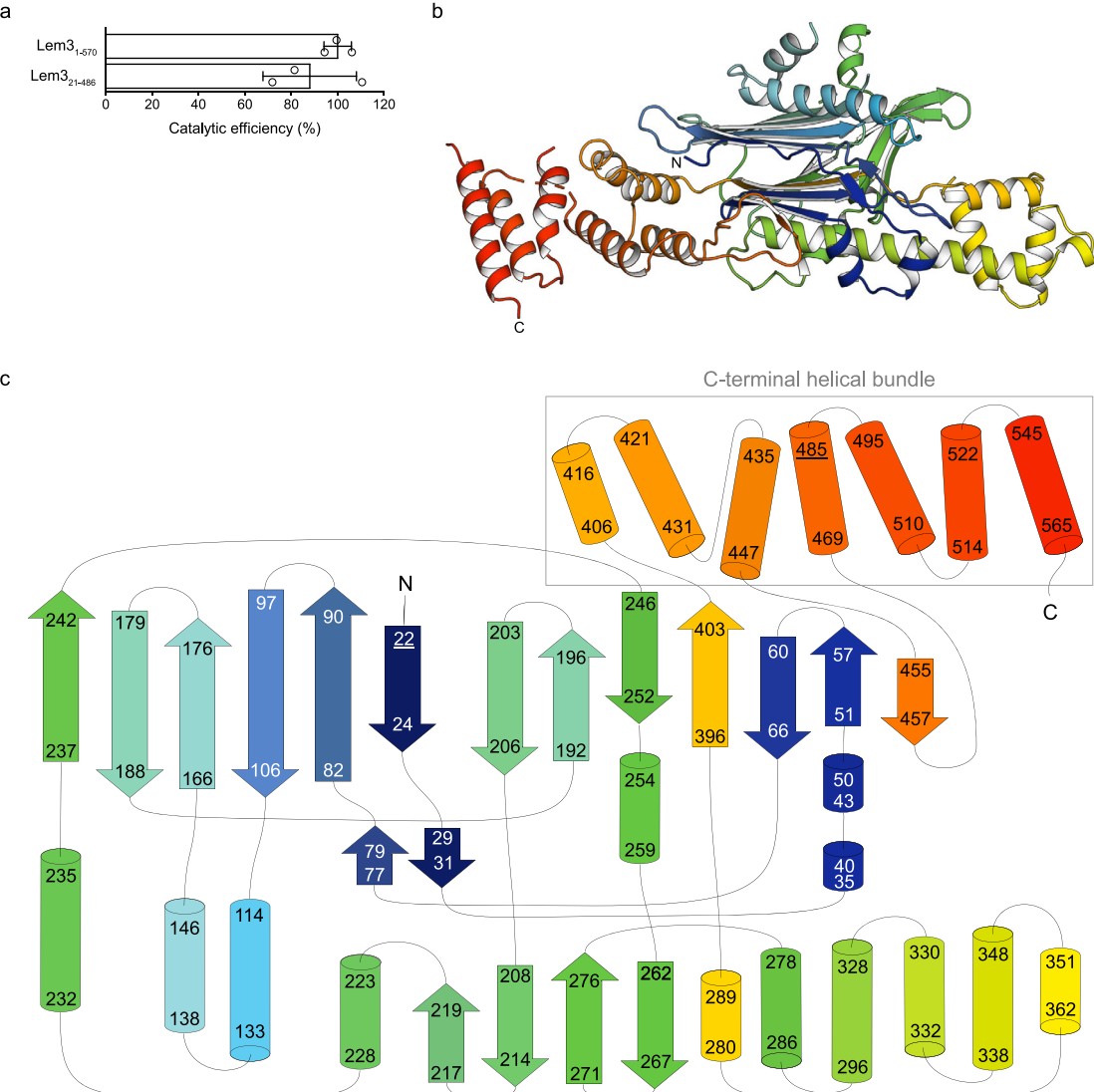

**Fig. 1 | Crystal structure of Lem3. a** Activity assay for full-length Lem3 and shortened construct. Catalytic efficiencies ($k_{cat}/K_M$) were determined from mass spectrometry-derived dephosphocholination curves. Means (±SD) represent three independent biological replicates (unpaired, two-tailed $t$-test; $p$ value: 0.38). **b** Cartoon representation of $Lem3_{FL}$ (coloured as rainbow from N- to C-terminus) (PDB ID: 8AGG). **c** Schematic representation of $Lem3_{FL}$ secondary structural motifs (numbers indicate amino acids contributing to the respective secondary structural motif). Colouring matches $Lem3_{FL}$ in (**b**).

## $Mg^{2+}$ and $Mn^{2+}$ increase Lem3 stability and activity

The presence of metal ions in the crystal structure and the structural homology to PPM phosphatases suggests that Lem3 shares a catalytic mechanism with this protein family. Therefore, we investigated the dependence of $Lem3_{21-486}$-mediated dephosphocholination of Rab1b-phosphocholine ($Rab1b_{S76(PC)}$) on divalent cations such as $Mg^{2+}$, $Mn^{2+}$, and $Ca^{2+}$.

First, we analysed the influence of different divalent ions on the stability of both $Lem3_{21-486}$ and $Lem3_{FL}$ via thermal unfolding monitored by differential scanning fluorimetry (nanoDSF). The thermal stability of both Lem3 constructs increases by 8, 17 or 11 °C in presence of $Mg^{2+}$, $Mn^{2+}$, or $Ca^{2+}$, respectively (Fig. 3a). Next, we studied the effect of $Mg^{2+}$, $Mn^{2+}$, or $Ca^{2+}$ on $Lem3_{21-486}$-mediated dephosphocholination of $Rab1b_{S76(PC)}$. Using intact mass spectrometry (MS), time-dependent $Rab1b_{S76(PC)}$ demodification by $Lem3_{21-486}$ was quantified. In the presence of $Ca^{2+}$, no dephosphocholination was observed. In contrast, $Mn^{2+}$ or $Mg^{2+}$ promote $Rab1b_{S76(PC)}$ dephosphocholination, with $Mn^{2+}$ having the strongest effect on activity (Fig. 3b).

In order to investigate the relevance of $D82_{Lem3}$, $D105_{Lem3}$, $D254_{Lem3}$ and $D394_{Lem3}$ for metal binding and dephosphocholination,

we compared the catalytic activities of the respective alanine-substituted $Lem3_{21-486}$-mutants (Fig. 3c). Additionally, the effect of the mutation of $D190_{Lem3}$ was examined since it is located in close proximity to the metal ions, suggesting that it could be involved in their coordination. Indeed, monitoring of $Rab1b_{S76(PC)}$ dephosphocholination by intact MS demonstrated that alanine substitution of these aspartates substantially decreased the catalytic activity by 75–97% relative to the wild-type protein (Fig. 3d). Therefore, metal ion binding is essential for the catalytic mechanism of $Rab1b_{S76(PC)}$ dephosphocholination.

## Lem3 can catalyse dephosphorylation and dephosphocholination

Lem3 shares a conserved structure with the PPM phosphatase superfamily. We, therefore, analysed its ability to also remove other phosphate-based modifications (Fig. 4a). First, we exploited the ability of the *Legionella* phosphocholine transferase AnkX to catalyse phosphate (P) or phosphoethanolamine (PE) attachment to S76 of Rab1b in vitro[32]. Preparative modification of Rab1b with phosphate or phosphoethanolamine was performed using AnkX in the presence of the

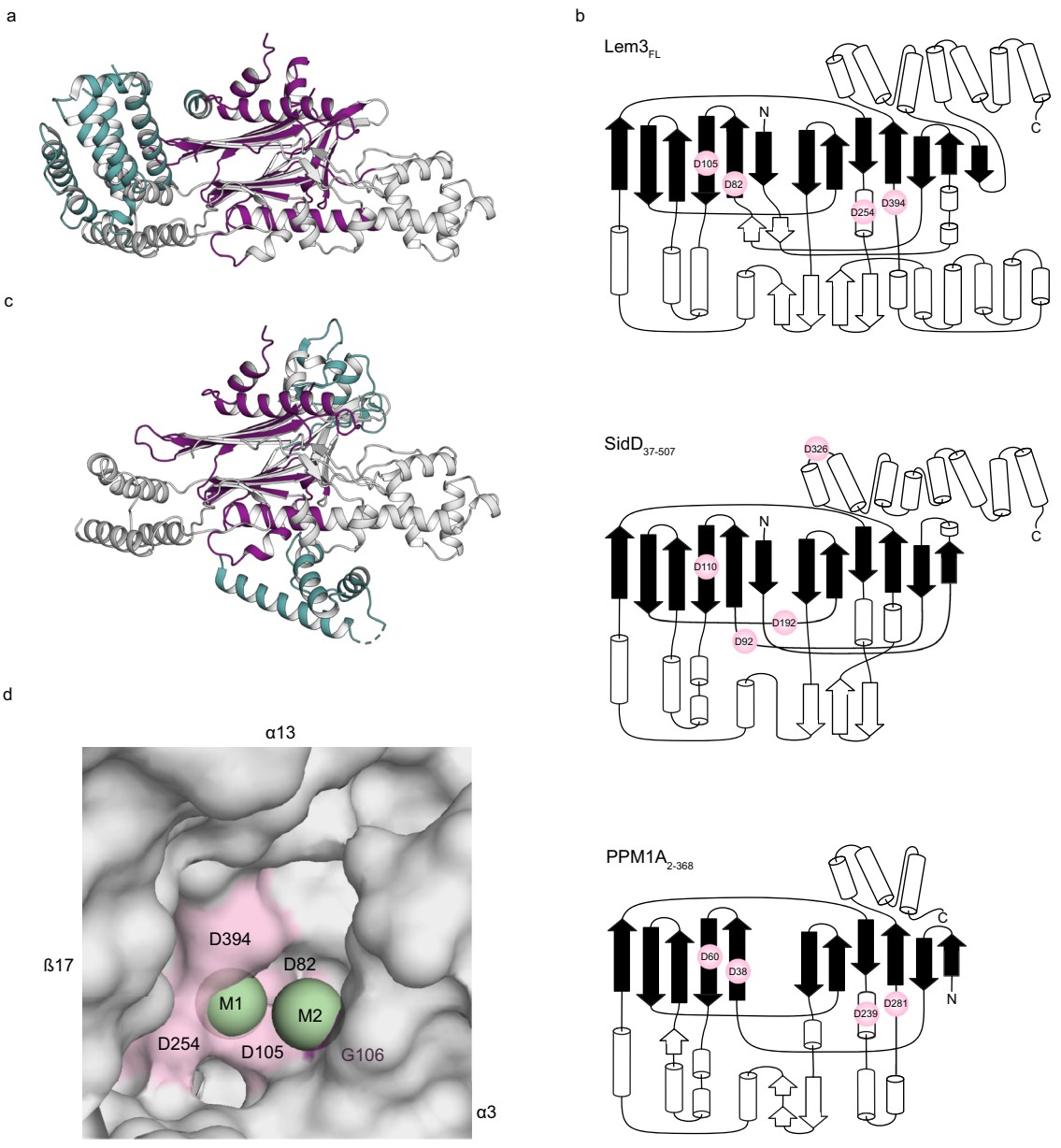

**Fig. 2 | Structural relationship of Lem3 to PPM phosphatases. a** Superimposition of Lem3$_{21-486}$ (Grey) (PDB ID: 8ANP) and SidD (PDB ID: 6RRE) (Turquoise) resulting in an RMSD value of 2.51 Å on 246 Cα-atoms superimposed. Secondary structural motifs shared by the two proteins are coloured in magenta. **b** Schematic representation of Lem3$_{21-486}$, SidD$_{39-507}$ and PPM1A$_{2-368}$. β-strands contributing to the central β-sheets are coloured in black, positions of aspartate residues coordinating metal ions are shown as spheres (Pink). **c** Superimposition of Lem3$_{21-486}$ (PDB ID: 8ANP) (Grey) and PPM1A (PDB ID: 4RA2) (Turquoise) resulting in an RMSD value of 3.53 Å on 171 superimposed Cα-atoms. Secondary structural motifs shared by the two proteins are coloured in magenta. **d** Surface representation of Lem3's catalytic pocket. Metal ions are represented as spheres (Green) and locations of coordinating residues are highlighted in pink (aspartate) and magenta (glycine).

co-substrates CDP and CDP-ethanolamine, respectively, and validated using intact MS (Supplementary Fig. 2a).

Demodification by Lem3$_{21-486}$ was monitored using intact MS (Fig. 4b). Additionally, dephosphorylation was checked by Phos-Tag SDS-PAGE (sodium dodecylsulfate polyacrylamide gel electrophoresis), based on the decreased migration of phosphorylated proteins (Fig. 4c). The catalytic efficiency of Lem3$_{21-486}$ for Rab1b$_{S76(PE)}$ and Rab1b$_{S76(PC)}$ showed no significant difference (Fig. 4b). Full demodification of Rab1b$_{S76(PC)}$ and Rab1b$_{S76(PE)}$ by Lem3$_{21-486}$ was observed after overnight incubation. However, demodification of Rab1b$_{S76(P)}$ takes substantially longer compared to Rab1b$_{S76(PC)}$ and Rab1b$_{S76(PE)}$. Only stoichiometric quantities of Lem3$_{21-486}$ lead to full dephosphorylation of Rab1b$_{S76(P)}$ within 24 hours (monitored by Phos-Tag SDS-PAGE, Fig. 4c). Thus, Lem3 is a poor phosphatase

despite the apparently high structural similarity in and around the active centre.

We wondered whether other Rab1b PTMs in the vicinity of S76 may affect Lem3-mediated dephosphocholination. Therefore, Rab1b was modified with PC at S76 and AMP at Y77 (Rab1b$_{S76(PC)Y77(AMP)}$). During Legionella infection, human Rab1b is modified by the Legionella effector DrrA at Y77 with an AMP moiety[33]. We observed full dephosphocholination of Rab1b$_{S76(PC)Y77(AMP)}$ after overnight incubation with Lem3$_{21-486}$. Despite the close proximity of the sterically demanding AMP-group, the rate of dephosphocholination of Rab1b$_{S76(PC)Y77(AMP)}$ is only moderately impaired compared to Rab1b$_{S76(PC)}$ (Fig. 4b).

In addition to Rab1b, also Rab35 can be phosphocholinated in vitro by AnkX. Rab1b and Rab35 are homologous proteins that share

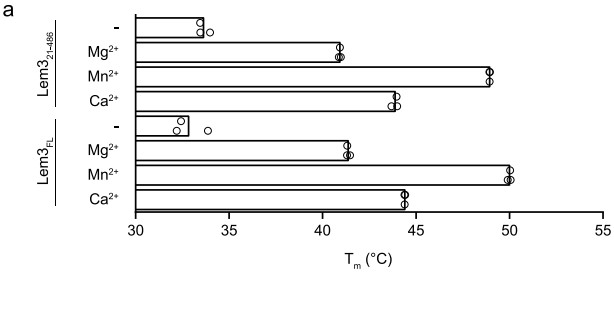

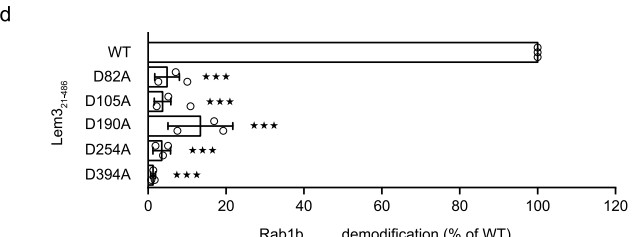

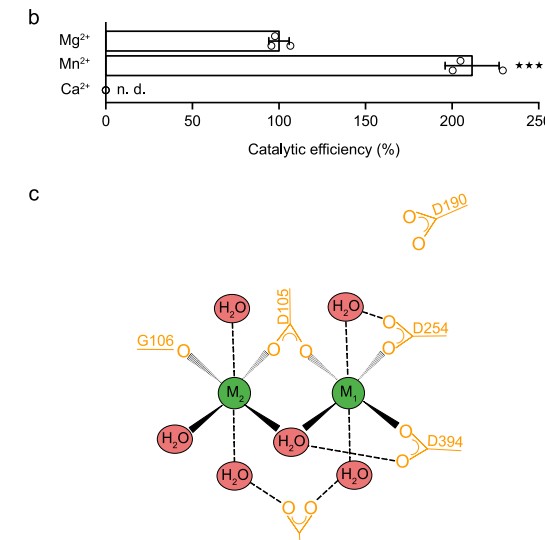

**Fig. 3 | Divalent cation dependence of Lem3 in structural integrity and catalysis. a** NanoDSF measurements of $Lem3_{21-486}$ and $Lem3_{FL}$ without or in presence of $Mg^{2+}$, $Mn^{2+}$ or $Ca^{2+}$. The melting temperatures of $Lem3_{21-486}$ and $Lem3_{FL}$ are not visibly different, indicating that the metal ions bind and stabilise mainly the core domain of Lem3. **b** Catalytic efficiencies of $Lem3_{21-486}$ in presence of different divalent cations. Catalytic efficiencies were determined from MS-derived dephosphocholination curves. Means (±SD) represent three independent biological replicates (unpaired, two-tailed $t$-test; $p$ value: 0.0003 (***)).n.d. (not determinable) (**c**) Schematic depiction of metal ion ($M_1$ and $M_2$) coordination by Lem3. **d** Activity assay for Lem3 alanine mutants of metal ion coordinating aspartate residues. Percentage of $Rab1b_{S76(PC)}$ demodification was monitored after 24 h of equimolar co-incubation. Means (±SD) represent three independent biological replicates (unpaired, two-tailed $t$-test; all $p$ values equal less than 0.0001(***)).

81% amino acid sequence identity (Supplementary Fig. 3a). Furthermore, they are virtually structurally identical, since superimposition of Rab1b and Rab35 results in a RMSD of 0.915 Å on 164 superimposed Cα-atoms (Supplementary Fig. 3b). Although Rab35 is phosphocholinated at the homologous position, T76 is modified in Rab35 instead of the S76 observed in Rab1b. Interestingly, Lem3 cannot cleave the PC-group from threonine-containing Rab-variants (i.e. $Rab1b_{S76T(PC)}$ or $Rab35_{T76(PC)}$), but can dephosphocholinate serine-containing Rabs (i.e. $Rab1b_{S76(PC)}$ or $Rab35_{T76S(PC)}$): Neither $Rab1b_{S76T(PC)}$ nor $Rab35_{T76(PC)}$ can be dephosphocholinated within 24 h, while $Rab1b_{S76(PC)}$ and $Rab35_{T76S(PC)}$ are demodified at comparable rates by $Lem3_{21-486}$ (Fig. 4d). In conclusion, Lem3 is able to remove different phosphomonoester or phosphodiester linkages of serine but not threonine residues.

## Site-specific cross-linking of Lem3:Rab1b complexes

The structure and functional analysis of $Lem3_{21-486}$ provides a rationale for the catalytic mechanism reminiscent of PPM phosphatases. To further characterise the structural basis for $Rab1b_{S76(PC)}$ recognition, we attempted to obtain its complex structure with $Lem3_{21-486}$. Due to the transient nature of their interaction, co-crystallisation of the proteins was unsuccessful. Therefore, we used a site-specific covalently linking strategy to capture the Lem3:Rab1b complex. We applied a previously established procedure that uses a CDP-choline derivative bearing a thiol-reactive chloroacetamide moiety at the quaternary ammonium linked via a C3 linker (referred to as CDP-choline-Cl) (Fig. 5a)[20,34] to covalently link the minimal GTPase-domain of Rab1b (aa 3-174) and $Lem3_{21-486}$. First, AnkX was used to quantitatively modify Rab1b with phosphocholine-chloroacetamide (PC-Cl) at S76 ($Rab1b_{S76(PC-Cl)}$). Next, we incubated $Rab1b_{S76(PC-Cl)}$ with $Lem3_{21-486}$ for covalent complex formation. $Lem3_{21-486}$ contains five endogenous cysteines of which two $C209_{Lem3}$ and $C394_{Lem3}$ are located in proximity to the catalytic centre (i.e. the metal ion binding site). In order to reduce spontaneous cleavage of the phosphodiester linkage by Lem3, we additionally introduced the $D190A_{Lem3}$ mutation that substantially decreased hydrolytic activity but did not affect purification yields

(Fig. 3d). A covalent $Lem3_{21-486, D190A}$: $Rab1b_{S76(PC-Cl)}$ complex readily formed as indicated by an increase in apparent molecular weight in SDS-PAGE (Fig. 5b). The presence of multiple species is likely due to unspecific reaction with the Lem3 cysteine. Successive cysteine substitutions revealed that the construct $Lem3_{D190A\_C134S\_C209S\_C456S}$ formed a single covalent species with $Rab1b_{S76(PC-Cl)}$ linked via $Cys395_{Lem3}$ (Fig. 5c). Since the mutation $D190A_{Lem3}$ was not sufficient to prevent cleavage of the phosphodiester, we use the S76T-substitution in Rab1b ($Rab1b_{S76T(PC-Cl)}$) for further complex formation. This allowed us to exploit the inability of Lem3 to cleave the phosphate at threonine residues, thereby gaining an hydrolysis-deficient complex. Also, the linkage via $Cys395_{Lem3}$ may result in suboptimal positioning of Rab1b in the complex interface, as indicated by low overall complex yields. Therefore, we introduced further cysteine substitutions in the putative $Lem3_{21-486}$ active centre in order to improve yields (Fig. 5d). Indeed, the substitution $T391C_{Lem3}$ readily produced preparative quantities of the complex $Lem3_{T391C}$:$Rab1b_{S76T(PC-C3)}$ and $Lem3_{T391C}$:$Rab35_{T76(PC-C3)}$ as demonstrated by SDS-PAGE (Fig. 5e, Supplementary Fig. 3c, Supplementary Fig. 4a, b). Size exclusion chromatography and SDS-PAGE gel shift analysis demonstrated that the purified complex $Lem3_{C134S\_C209S\_T391C\_C395S\_C456S}$:$Rab1b_{S76T(PC-C3)}$ (referred to as complex$_{T391C}$) was present as a pure monomeric species (Fig. 5f). Intact MS confirmed the complex purity and identity (Fig. 5g). Even though the catalytic activities of the cysteine substitution mutant ($Lem3_{C134S\_C209S\_C395S\_C456S}$) and the crystallisation construct ($Lem3_{C134S\_C209S\_T391C\_C395S\_C456S}$) are substantially decreased by >99% (Supplementary Fig. 4c, d) compared to wild type, their tertiary structures are identical as outlined below. Thus, the mutations do not affect the overall structure of Lem3 or the arrangement of amino acids within the active site, although minor distortions may occur that cause reduction of activity. Furthermore, as demonstrated for Lem3 wild-type, the mutants cannot demodify $Rab1b_{S76T(PC)}$ (Supplementary Fig. 4e), permitting us to obtain a stable complex. Crystals of complex$_{T391C}$ diffracted to 2.15 Å and hence permitted its structure determination.

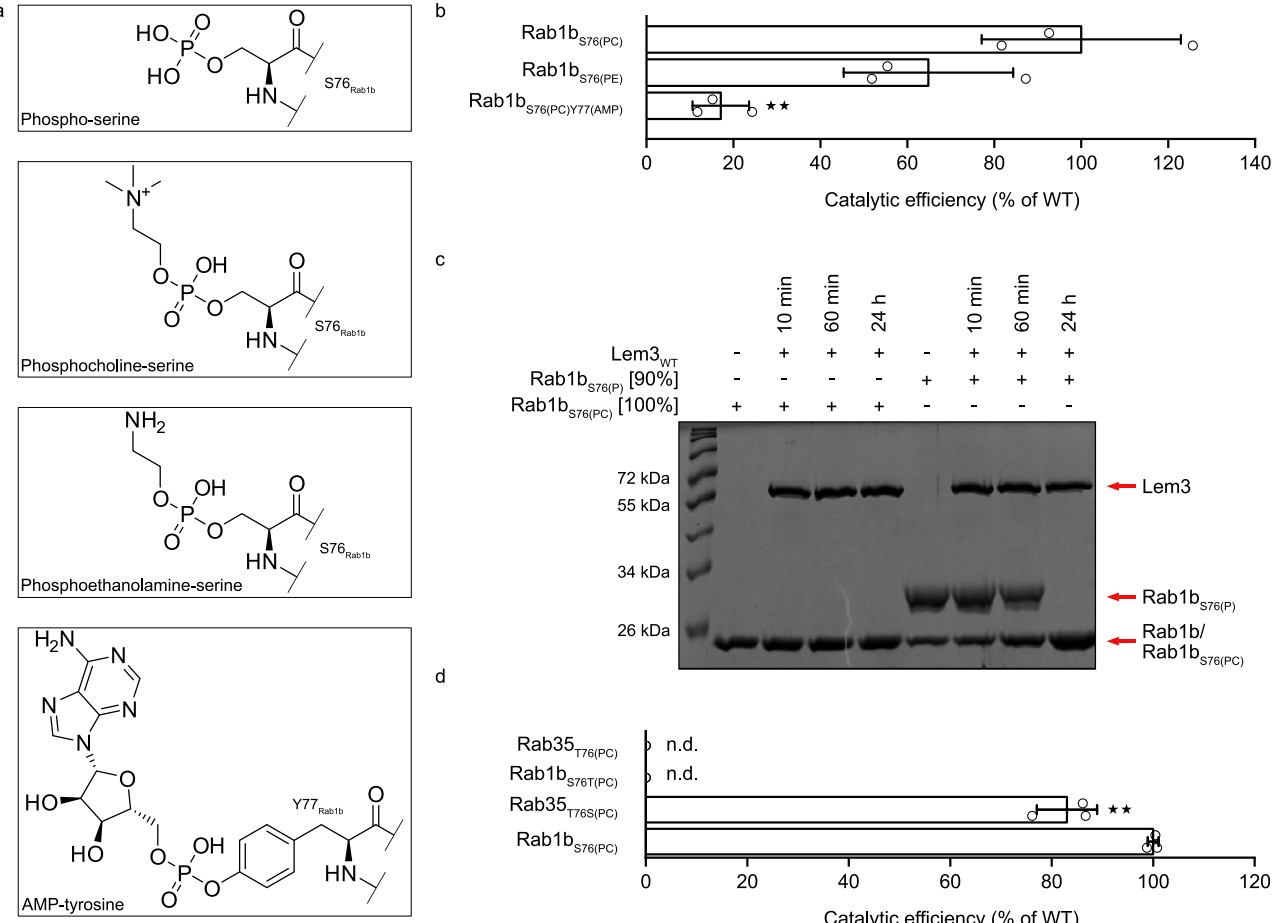

**Fig. 4 | Enzymatic analysis of Lem3. a** Structures of post-translationally modified amino acids of Rab1b used for enzymatic analysis of Lem3. **b** Catalytic efficiencies of Lem3$_{21-486}$ for Rab1b$_{S76(PC)}$, Rab1b$_{S76(PE)}$ and Rab1b$_{S76(PC)Y77(AMP)}$. Catalytic efficiencies were determined from MS-derived dephosphocholination curves. Means (±SD) represent three independent biological replicates (unpaired, two-tailed $t$-test; $p$ values equal 0.004(**) and 0.11). **c** Demodification of Rab1b$_{S76(PC)}$/Rab1b$_{S76(P)}$ by Lem3$_{21-486}$ resolved by Phos-Tag™ SDS-PAGE. Proteins were

incubated at equimolar ratios. Samples were taken at indicated time points. 91% of the Rab1b$_{S76(P)}$ sample is modified. Red arrows indicate band identity. **d** Catalytic efficiency of Lem3$_{21-486}$ on Rab1b$_{S76(PC)}$/Rab35$_{T76(PC)}$ and Rab1b$_{S76T(PC)}$ or Rab35$_{T76S(PC)}$ mutants. Catalytic efficiencies were determined from MS-derived dephosphocholination curves. Means (±SD) represent three independent biological replicates (unpaired, two-tailed $t$-test; $p$ value equals 0.009(**)).

## Structure of the Lem3:Rab1b complex$_{T391C}$

The complex crystal structure was solved by molecular replacement using the structures of Rab1b$_{3-174}$ (PDB ID: 3NKV)[12] and the Lem3$_{21-486}$ (PDB ID: 8ANP) structure as search models (Fig. 6a, b, Supplementary Table 1). The electron density allowed the construction of the protein chains at a high level of details, in particular at the interface where density was observed for the connected amino acids T76$_{Rab1b}$ and C391$_{Lem3}$ and the PC-C3-group (Fig. 6c). Nevertheless, weak electron density and the high B-factors for the linker atoms indicate structural flexibility (Supplementary Fig. 5a). Therefore, the covalent linker likely does not force Rab1b and Lem3 into artificial conformations (Supplementary Fig. 5b, unbiased map). No specific interactions are visible for the C3 linker region while L68$_{Lem3}$ shows hydrophobic interaction with the choline group (Fig. 6c, d). The position of L68$_{Lem3}$ is occupied by an arginine (or the functionally related lysine) in most PPM phosphatases and by a phenylalanine in SidD (Supplementary Fig. 5c, d). The side chain of R33$_{PPM1A}$ is known to be involved in coordination of the substrates phosphate group[35] (Supplementary Fig. 5e). Amino acid F74$_{SidD}$ is supposed to interact with the adenine base of the AMP moiety attached to Y77 in AMPylated Rab1b, since F74$_{SidD}$ substitution with alanine strongly affects Rab1b-AMP deAMPylation[36].

Superimposition of the complex$_{T391C}$ and the Rab1b-unbound Lem3 revealed a second amino acid potentially involved in coordination

of the choline group. Due to the modification of T391$_{Lem3}$ to cysteine for covalent linkage, no interactions can be observed in the complex structure itself, but in silico replacement of C391$_{Lem3}$ by the natural threonine hints at hydrophobic interaction between the methyl groups of the choline group and T391$_{Lem3}$. Together, L68$_{Lem3}$ and T391$_{Lem3}$ allow the formation of a hydrophobic environment which favours the proper accommodation of the choline group in the catalytic site.

The phosphate moiety of the linker in complex$_{T391C}$ does not participate in direct interactions with amino acids, but is involved in the coordination of three metal ions located in the active centre of Lem3 at the same site as observed in the Lem3$_{21-486}$ structure and coordinated by the same residues (D105$_{Lem3}$, D254$_{Lem3}$, and D394$_{Lem3}$, D105$_{Lem3}$ and G106$_{Lem3}$) (Fig. 6d). Notably, in the coordination of M2, a water molecule is substituted by an oxygen atom provided by the phosphodiesters group of the phosphocholine moiety. In addition, a third metal ion (M3) can be placed in the electron density, which is coordinated by D254$_{Lem3}$ and D190$_{Lem3}$ (Supplementary Fig. 5e). Water molecules complete the metal coordination at expected positions. Substitution by an alanine of D190$_{Lem3}$ has a similar deleterious effect of Lem3 catalytic efficiency as the mutation of the aspartate residues coordinating M1 and M2, suggesting that the presence of a metal ion in position M3 is essential for the proper dephosphocholination by Lem3 (Fig. 3c).

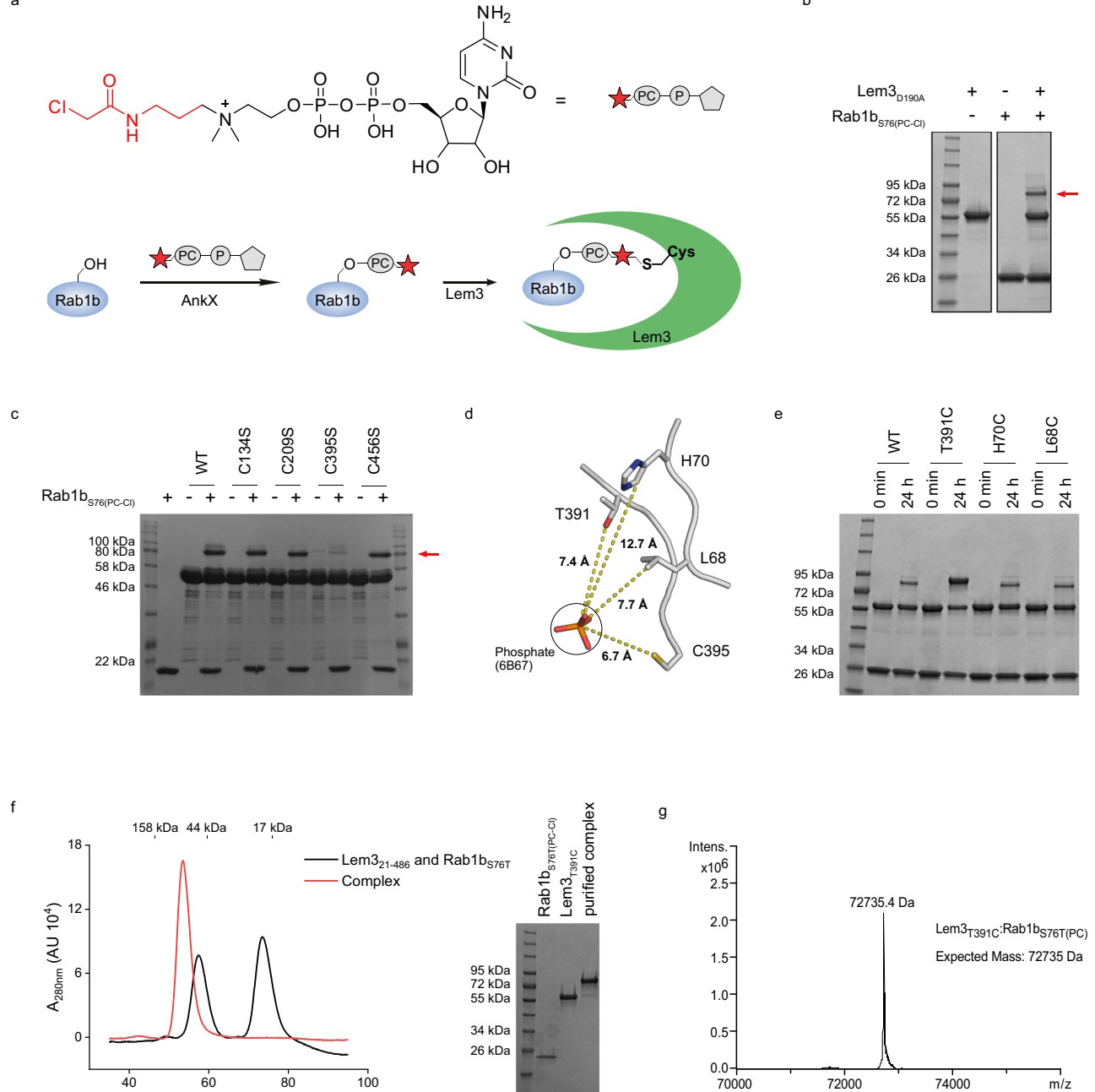

**Fig. 5 | Covalent complex formation. a** Strategy for covalent complex formation between Lem3 and Rab1b. Natural CDP-choline (Black) was equipped with a thiol-reactive chloroacetamide functionality separated from the choline group with a C3 linker (Red). **b** Complex formation between Lem3 and Rab1b$_{S76(PC-Cl)}$ monitored by SDS-PAGE Gel shift assay. The red arrow indicates the band for the specific complex. Samples were run on the same gel, black frame indicates cut. **c** Analysis of unspecific complex formation between endogenous Lem3 cysteine and Rab1b$_{S76(PC-Cl)}$ monitored by SDS-PAGE Gel shift assay. The red arrow indicates the band for the specific complex. **d** Lem3 residues selected for cysteine mutations in distance to phosphate as coordinated by PPM1A (PDB ID: 6B67) (dashed yellow lines) when Lem3 (PDB ID: 8ANP) and PPM1A are superimposed. γ-C atoms were used for distance measurements given in Ångström. **e** Evaluation of covalent complex formation between Rab1b$_{S76T(PC-Cl)}$ and Lem3/Lem3$_{Cys}$. Covalent complex formation was quantified using SDS-PAGE Gel shift assay. **f** Overlay of size exclusion chromatograms of mixed Lem3$_{T391C}$ and Rab1b$_{S76T}$ and complex$_{T391C}$. SDS-PAGE shows input on the size exclusion column. A$_{280nm}$, absorbance at 280 nm. **g** Intact MS of complex$_{T391C}$ before crystallisation.

Upon binding to its protein substrate Rab1b, Lem3$_{21-486}$ does undergo only minor conformational changes (Supplementary Fig. 6a). However, a new β-strand is formed by the α12-α13$_{Lem3}$ loop and a slight lever movement of the four α-helices constituting the thumb (α9-α13) is observed, leading to broadening of the hollow region (located at the base of the core β-sheet cleft between fist and thumb). The core structure of Rab1b$_{3-174}$ in the complex is virtually identical to unbound Rab1b (Fig. 6e). Indeed, Rab1b maintains the typical GTPase fold with

β1-β6$_{Rab1b}$ forming a central six-stranded β-sheet surrounded by five α-helices (α1-α5$_{Rab1b}$). Only the region encompassing the switch region II (A67$_{Rab1b}$-G81$_{Rab1b}$) of Rab1b shows substantial structural rearrangement upon complex formation (Fig. 6e).

Rab1b:GDP binds to the hollow region of Lem3$_{21-486}$. The β-hairpin β12-β13$_{Lem3}$ of Lem3 protrudes into the Rab1b-region formed by α3$_{Rab1b}$ and switch II. As a consequence, the sequence R71-T74$_{Rab1b}$ of switch II is reorganized into a β-strand (β'$_{β3-4,Rab1b}$) and placed between

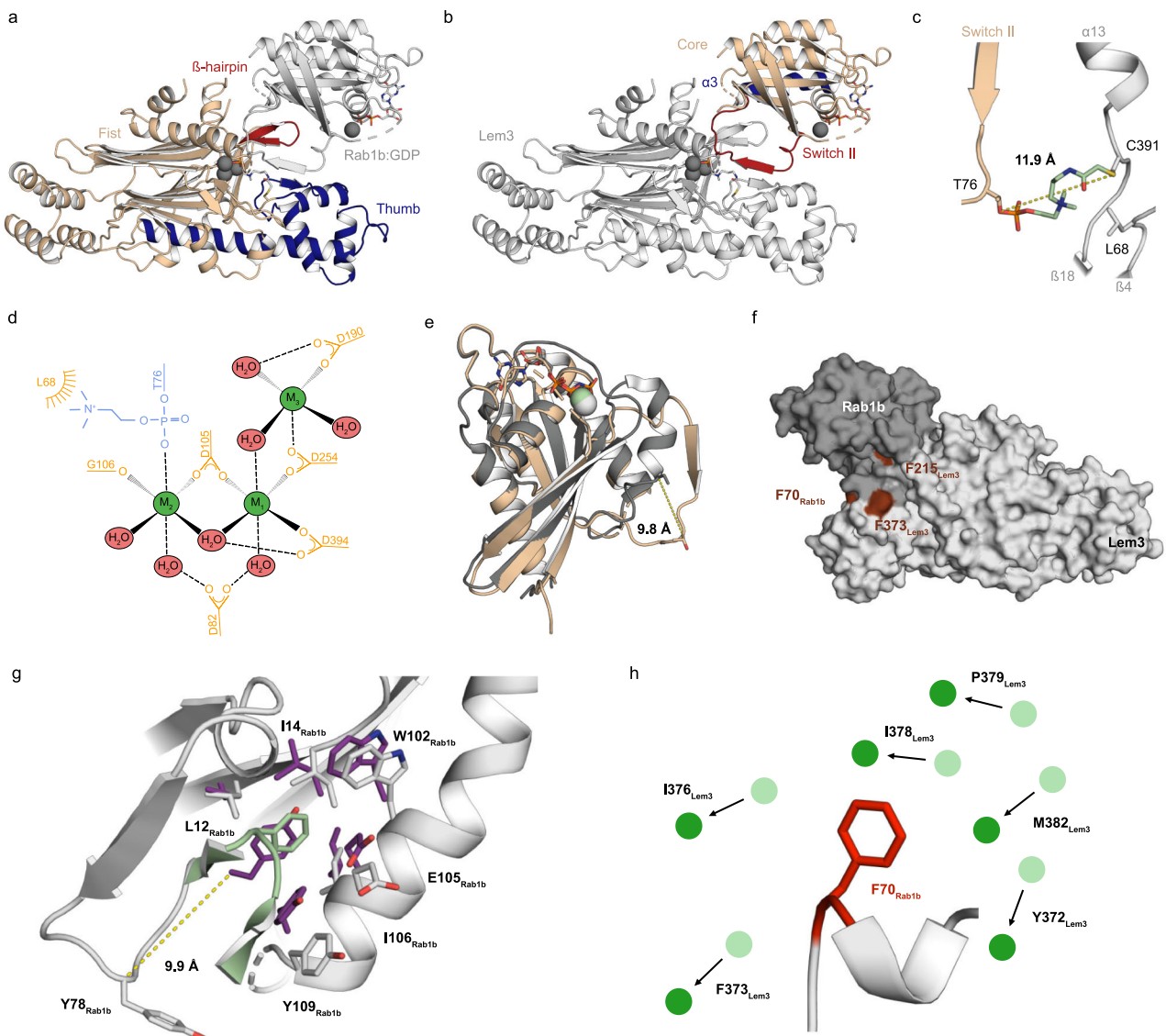

**Fig. 6 | Structure of the Lem3:Rab1b complex$_{T391C}$. a, b** Cartoon representation of the crosslinked complex$_{T391C}$ (PDB ID: 8ALK). The GDP, the linker, T76$_{Rab1b}$ and C391$_{Lem3}$ are represented as sticks. Metal ions are represented as spheres (dark grey). **c** Cartoon representation of the crosslinked region in complex$_{T391C}$. The linker (Green, coloured by atoms), T76$_{Rab1b}$ (Wheat), L68$_{Lem3}$ and C391$_{Lem3}$ (Grey) are represented as sticks. Dashed line indicates the distance between the γ-atoms of T76$_{Rab1b}$ and C391$_{Lem3}$ given in Ångström. **d** Schematic depiction of metal ion (M$_1$ - M$_3$) coordination in the catalytic centre of Lem3 in complex$_{T391C}$ (**e**) Super-imposition of Rab1b (PDB ID: 3NKV) (Grey) and Lem3-bound Rab1b (Wheat) (PDB ID: 8ALK), S76$_{Rab1b}$ and T76$_{Rab1b}$ are represented as sticks and movement in space induced by Lem3 is indicated as dashed line and measured in Ångström. **f** Surface representation of complex$_{T391C}$. Locations of hydrophobic patches are highlighted in red. **g** Cartoon representation of Lem3 bound Rab1b (Grey) and Lem3 thorn with F215$_{Lem3}$ at the tip (Green). Interacting amino acids are shown as sticks. Corresponding Rab1b amino acids of unbound Rab1b (PDB ID: 3NKV) (Purple) are shown as sticks. Movement in space induced by Lem3 is indicated as dashed line and measured in Ångström. **h** Cartoon representation of F70$_{Rab1b}$ in complex$_{T391C}$. Replacement of amino acids of Lem3 by F70A$_{Rab1b}$ indicated by black arrows. Positions of Lem3 amino acids in the Lem3$_{21-486}$ structure are shown as spheres (light green) and their corresponding position in the complex structure is shown as sphere (dark green).

the β-hairpin (located on top of the fist) and the thumb. To make room for the β'$_{β3-4,Rab1b}$, the cleft between thumb and fist opens up, indicated by a 4 Å movement of E381$_{Lem3}$. Additionally, the new β-strand (β'$_{β14-β15,Lem3}$) forms in the thumb out of a loop region (I370-Y372$_{Lem3}$). Together with the already present β-strands (β12-13$_{Lem3}$ and β16-17$_{Lem3}$), it forms a new 6-stranded intermolecular β-sheet (Supplementary Fig. 6b). As a consequence, the switch II region is displaced from the Rab1b core. Also, the phosphocholinated T76$_{Rab1b}$ moves towards the active centre of Lem3, resulting in close proximity of the phosphodiester group to the coordinated metal ion site (Fig. 6a–c, e).

The complex interface is characterised by three major hydrophobic patches (referred to as HP$_{I-III}$) (Fig. 6f). Polar interactions are

mainly limited to the β-strand hydrogen (H-)bonds (Supplementary Fig. 6c, d). Two phenylalanine residues from Lem3 and one from Rab1b form the basis for all three hydrophobic patches. Amino acid F215$_{Lem3}$ (HP$_I$) is located on top of the β-hairpin and positions between α3 and switch II of Rab1b by interacting with the hydrophobic core of Rab1b (Fig. 6g), formed by L12$_{Rab1b}$, I14$_{Rab1b}$, Y78$_{Rab1b}$, W102$_{Rab1b}$, E105$_{Rab1b}$, I106$_{Rab1b}$ and Y109$_{Rab1b}$. F215$_{Lem3}$ occupies the position of Y78$_{Rab1b}$, which is located in the hydrophobic pocket in the Rab1b-apo-structure. As a consequence, Y78$_{Rab1b}$ is displaced by 9.9 Å (distance between Cα-atoms), likely contributing to the movement of switch II towards Lem3's active centre. This enables the phosphocholinated T76$_{Rab1b}$ to approach the metal ions and thus the active centre in Lem3. HP$_{II}$ is

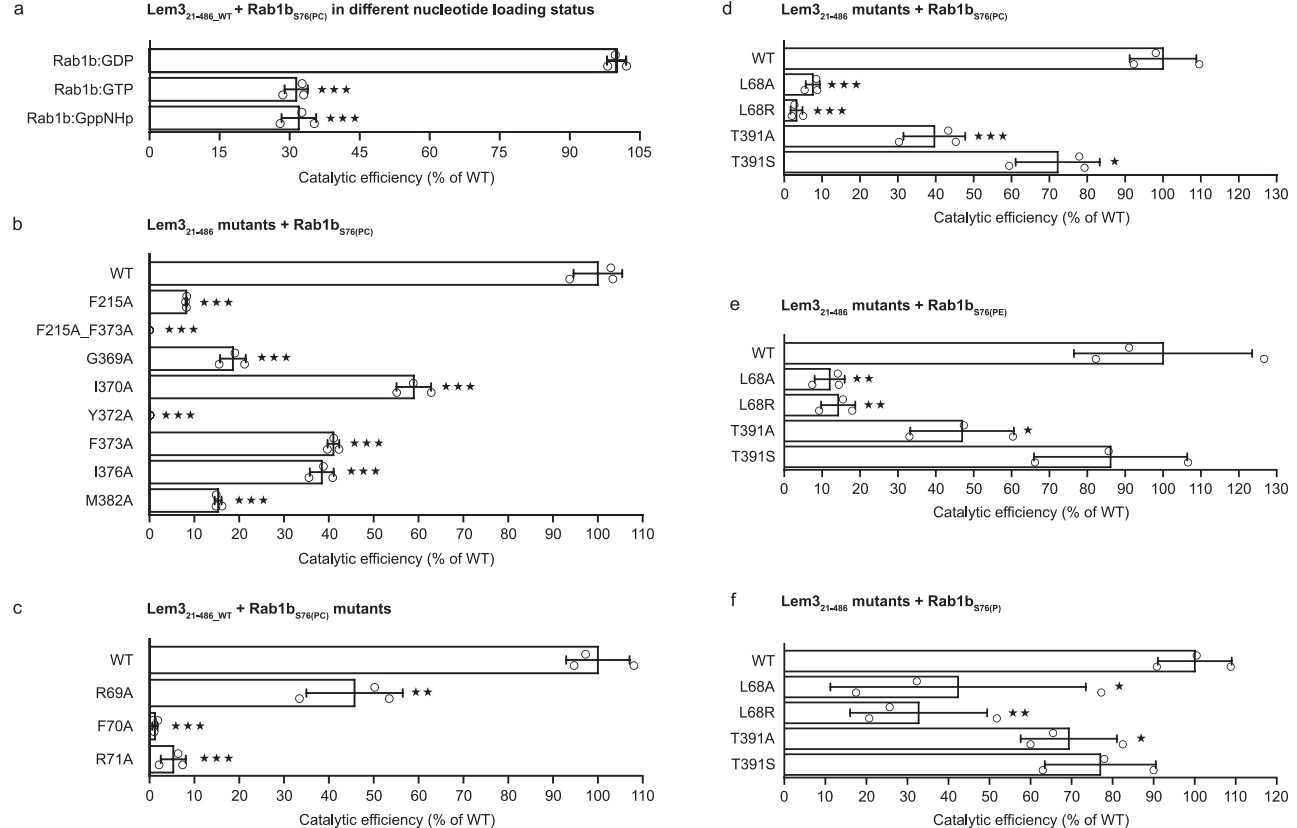

**Fig. 7 | Catalytic efficiency of Lem3. a** Catalytic efficiencies depending on nucleotide-bound status of Rab1b. Means (±SD) represent three independent biological replicates (unpaired, two-tailed $t$-test; all $p$ values equal less than 0.0001(***)). **b** Catalytic efficiencies of Lem3$_{21-486}$ mutants involved in protein-protein interactions. Means (±SD) represent three independent biological replicates (unpaired, two-tailed $t$-test; all $p$ values equal less than 0.0001 except for Lem3$_{I370A}$ with $p = 0.0004$ (all ***)). **c** Catalytic efficiencies for Rab1b mutants. Means (±SD) represent three independent biological replicates (unpaired, two-tailed $t$-test; all $p$ values equal less than 0.0001 (***) except for Rab1b$_{R69A}$ with 0.0018 (**)). **d–f** Catalytic efficiencies for Lem3$_{21-486}$ mutants involved in choline group (of phosphocholine) coordination on Rab1b$_{S76(PC)}$ (**d**), Rab1b$_{S76(PE)}$ (**e**) and Rab1b$_{S76(P)}$ (**f**). Means (±SD) represent three independent biological replicates (unpaired, two-tailed $t$-test). $P$ values from top to bottom for (**d**) (less than 0.0001, less than 0.0001, 0.001 (all ***), 0.0289(*)) for (**e**) (0.0032, 0.0036 (both **), 0.0279 (*), 0.4932) and for (**f**) (0.0359 (*), 0.0035 (**), 0.0236 (*), 0.0678).

formed by F373$_{Lem3}$ located N-terminal to the newly formed β-strand β'$_{β14-β15,Lem3}$ in Lem3. The residue interacts with the residues N-terminal of the switch II region (R69-T72$_{Rab1b}$) of Rab1b.

HP$_{III}$ is generated by F70$_{Rab1b}$. HP$_{II}$ and HP$_{III}$ overlap with each other as F70$_{Rab1b}$ and F373$_{Lem3}$ interact via a β-strand H-bond. F70$_{Rab1b}$ binds into a pocket comprising Y372$_{Lem3}$, I376$_{Lem3}$, I378$_{Lem3}$, P379$_{Lem3}$ and M382$_{Lem3}$. The amino acids of Lem3 involved in this pocket are displaced by 2.9-3.9 Å by F70$_{Rab1b}$ (Fig. 6h). Since Lem3 can also form complexes with Rab35, we investigated whether the key interacting residues of Rab1b are conserved in Rab35. Most amino acids are indeed conserved between Rab1 and Rab35, such as F70 (Supplementary Fig. 3b) and amino acids of the hydrophobic core (Supplementary Fig. 3d, e), with the exception of Y109$_{Rab1b}$, which is N109 in Rab35 (Supplementary Fig. 3d, e).

Taken together, it appears that the Rab1b switch II region encompassing amino acids F70-Y78$_{Rab1b}$ are held in a specific conformation via F70$_{Rab1b}$, Y70$_{Rab1b}$ and β'$_{β14-β15,Rab1b}$ interaction with Lem3.

## Dephosphocholination through local substrate remodelling

The switch II region undergoes structural remodelling upon complex formation with Lem3. Since the switch II conformation is determined by the nucleotide state of Rab1b, Lem3 may have a preference for the GDP- or GTP-bound form. Hence, the rates of Lem3-mediated dephosphocholination of GppNHp (Guanosine-5'-[(β,γ)-imido]triphosphate;

a non-hydrolysable GTP-derivative), GTP and GDP bound Rab1b$_{S76(PC)}$ was determined by intact MS. A significant decrease by approximately 30% in Lem3 catalytic efficiency was observed for Rab1b$_{S76(PC)}$ loaded with GTP (0.23 μM$^{-1}$ s$^{-1}$) or GppNHp (0.24 μM$^{-1}$ s$^{-1}$) in comparison to GDP (0.74 μM$^{-1}$ s$^{-1}$) (Fig. 7a). Thus, Rab1b$_{S76(PC)}$ is a preferred Lem3-substrate in the GDP-state where the switch II region is supposed to be conformational more flexible rather than in the active, GTP-bound state, in which switch II is conformationally restrained[6].

To further validate the complex crystal structure, we investigated the impact of selected alanine substitutions in HP$_{I-III}$ using MS-derived dephosphocholination kinetics. Comparison of the catalytic efficiencies of the substitutions F70A$_{Rab1b}$ (1%), F215A$_{Lem3}$ (8.2%), F373A$_{Lem3}$ (41%) and F215A_F373A$_{Lem3}$ (0%) in comparison to the WT reveal major contribution of these amino acids for Rab1b$_{S76(PC)}$ dephosphocholination (Fig. 7b, c). While Y372A$_{Lem3}$ (0.12%) of HP$_{III}$ results in almost no detectable dephosphocholination activity, M382A$_{Lem3}$ (15.3%) strongly and I376A$_{Lem3}$ (38.4%) moderately impair dephosphocholination (Fig. 7b). The catalytic efficiency for R69A$_{Rab1b}$ (45.7%) and R71A$_{Rab1b}$ (5.3%) (HP$_{II}$) significantly differs from the one of WT (Fig. 7c). Strong effects on catalytic efficiency of Lem3 were observed for G369A$_{Lem3}$ (18.6%) and I370A$_{Lem3}$ (59%), which are part of the new β-strand β'$_{β14-β15}$ in Lem3 (Fig. 7b). None of the mutants mentioned above shows any effect on protein stability in nanoDSF measurements (Supplementary Fig. 6e, f).

Finally, we investigated the role of $L68_{Lem3}$ and $T391_{Lem3}$ in accommodating the hydrophobic choline group of phosphocholine by mutation to alanine ($L68A_{Lem3}$, $T391A_{Lem3}$), arginine ($L68R_{Lem3}$) or serine ($T391S_{Lem3}$). We analysed their ability to demodify $Rab1b_{S76(PC)}$, $Rab1b_{S76(PE)}$ and $Rab1b_{S76(P)}$. All mutants, except $T391S_{Lem3}$, show significantly decreased catalytic efficiencies for demodification of $Rab1b_{S76(PC)}$, $Rab1b_{S76(PE)}$ and $Rab1b_{S76(P)}$ (Fig. 7d–f). $T391S_{Lem3}$ only shows significantly reduced rates for $Rab1b_{S76(PC)}$. Mutation of $L68_{Lem3}$ to alanine or arginine leads to a strong reduction in catalytic efficiency for dephosphocholination (92% or 97%) and demodification of $Rab1b_{S76(PE)}$ (88% or 86%). Dephosphorylation rates of $L68A_{Lem3}$ and $L68R_{Lem3}$ are reduced by (58% or 68%). $T391S_{Lem3}$ shows 28% reduced catalytic efficiency for dephosphocholination, 14% for demodification of $Rab1b_{S76(PE)}$ and 23% for dephosphorylation. $T391A_{Lem3}$ similarly reduces dephosphocholination, dephosphoethanolination and dephosphorylation by 60%, 53% and 30%, respectively. In contrast to the mutants tested to validate the protein-protein interaction interface and the substitution $T391S_{Lem3}$, $T391A_{Lem3}$, $L68A_{Lem3}$ and $L68R_{Lem3}$ clearly differ from the $WT_{Lem3}$ in protein stability. $L68R_{Lem3}$ results in a protein which is more thermal stable by 4 °C while the $L68A_{Lem3}$ mutation reduce the melting temperature of Lem3 by 7 °C and $T391A_{Lem3}$ by 4 °C. (Supplementary Fig. 6e).

In conclusion, Lem3 preferentially dephosphocholinates GDP-bound $Rab1b_{S76(PC)}$. The Rab1b switch II is an essential element in recognition by Lem3. Furthermore, an arginine residue which is important for phosphate binding in PPM-phosphatases is functionally replaced in Lem3 by a leucine to bind the triple-methylated quaternary ammonium of the choline group.

## Discussion

Here we present the structures of the Legionella dephosphocholinase Lem3 in its apo-form and in complex with its human substrate protein Rab1b. Lem3 shows high structural relationship to PPM phosphatases. In addition to SidD, it is the second Legionella effector to reverse a PTM on Rab1b during infection[15]. Therefore, the evolutionarily conserved PPM-fold may serve as a template for diverse reactions catalysed by this group of proteins. Examples from other pathogenic bacteria using the PPM-fold for effector proteins are from *Thermosynechococcus elongates* (tPhpA)[37], *Mycobacterium tuberculosis* (MtPstP)[38], *Staphylococcus aureus* (Stp1)[39] and *Pseudomonas aeruginosa* (PppA)[40].

Our covalent capture approach using thiol-reactive substrate derivatives may serve as a basis to analyse other PPM-phosphatase-like proteins, their substrate interaction, and the underlying catalytic mechanisms. Since there is few structural data available for the substrate recognition by PPM phosphatases and none for SidD our findings provide valuable insights into how phosphatases recognise their substrate.

Although catalysing the hydrolysis of a phosphodiester bond instead of a phosphomonoester, the typical PPM catalytic centre is highly conserved in Lem3 and hence a similar catalytic mechanism as proposed for PPM phosphatases is likely[25]. As shown for PPM1A, Lem3 binds two metal ions in its apo-state, while three metal ions are observed in presence of substrate[25]. Metal ion coordination is performed by a set of highly conserved aspartate residues. Although we crystallised a hydrolysis deficient complex due to the S76T mutation in Rab1b, the Lem3:Rab1b crystal structure can be used to analyse the active centre since it contains the active Lem3 variant. The Lem3:Rab1b crystal structure indicates a catalytic mechanism involving the three metal ions and two molecule of waters. We speculate that the water molecule in ion-dipole interaction with M1 and M2 performs a nucleophilic in an $S_N2@P$-type attack on the phosphorus atom[41]. M3 may act as a Lewis acid, activating another water molecule to donate a proton for the phosphate/phosphocholine leaving group[42] (Supplementary Fig. 5f). The molecular basis for Lem3 performing

dephosphocholination but struggling with dephosphorylation may partially be explained by the complex structure: $L68_{Lem3}$ and $T391_{Lem3}$ are making hydrophobic contacts to the choline group of the linked phosphocholine moiety (Fig. 6d). In PPM phosphatases, this position is highly conserved and occupied by large positively charged amino acids ($R33_{PPM1A}$) (Supplementary Fig. 5c). Apparently, the hydrophobic interaction of the choline group with $L68_{Lem3}$ is important for dephosphocholination since mutation to alanine and arginine leads to strongly decreased activity compared to the WT (Fig. 7d). However, mutation to arginine does not convert Lem3 into a more efficient phosphatase although the mutation increases to protein stability. Substitution of $L68_{Lem3}$ with alanine, in contrast, results in a less stable protein, indicating an important functional role. Interestingly, the corresponding amino acid $F74_{SidD}$ in SidD was suggested to be involved in interaction with the AMP moiety of $Rab1b_{Y77(AMP)}$[36].

In addition to $L68_{Lem3}$, $T391_{Lem3}$ also contributes to the binding of the choline group. Therefore, we speculate that these positions in PPM like-proteins might contribute to the specificity towards different PTMs.

Unfortunately, the structure provides no explanation for the preference of Lem3 for serine-PC over threonine-PC. The biological relevance might be the discrimination between two Rab GTPases (Rab1b and Rab35) during the infection, both of them becoming modified by AnkX at serine or threonine, respectively[17]. Hence, AnkX may phosphocholinate Rab1 and Rab35, but only Rab1 is dephosphocholinated by Lem3.

In addition to the catalytic domain, we also provide a structure of the C-terminal part of Lem3, consisting of seven helices forming a bundle. Many PPM like-shaped proteins possess such C-terminal bundles, including SidD. A hydrophobic loop within the C-terminal helical bundle of SidD was shown to be responsible for membrane-localisation at the Golgi apparatus[36,43]. Since SidD also targets Rab1 during the infection we speculate that the C-terminal part of Lem3 might have a similar function and is responsible for Lem3-localization close to its target protein. However, a corresponding hydrophobic loop could not be identified in Lem3.

The crystal structure of the trapped Lem3:Rab1b complex provides insight into the interaction between a PPM phosphatase and its substrate protein. Although the hydrolysis-deficient complex structure cannot represent the active enzymatic state due to the inability of Lem3 to cleave threonine-PC-bonds, our data suggest that the complex structure represents a snapshot of the actual complex binding event of Rab1b by Lem3. So far, there is only one PPM in complex with a cyclic phosphorylated peptide available providing basic information on the orientation in the catalytic centre[25]. Our structure provides evidence that the individual regions of PPM phosphatases mediate substrate specificity. A structural comparison demonstrates, with one exception, that all analysed PPM phosphatases harbour a structurally conserved positively charged amino acid interacting with the phosphate group. In Lem3 and SidD, however, the equivalent position is occupied by the leucine or phenylalanine, which are responsible for the binding of the choline and AMP moieties, respectively[36]. Therefore, we suggest this position to be decisive for the catalytic specificity of the PPM like-shaped enzyme. Also our complex supports the hypothesis of a third metal ion being necessary for hydrolytic cleavage of the phosphodiester[25]. It also provides a methodical basis to trap and investigate PPM phosphatase substrate-protein complexes using corresponding ATP analogues, resulting in cysteine reactive phosphorylation of substrate proteins.

Although the Lem3-structure was correctly modelled by AF2, a complex structure prediction between Lem3 and Rab1b using AF2 failed. Presumably, the paucity of experimental atomic structures of transient protein complexes impairs the ability of AF2 to overcome those challenges. This argument also applies to the fact that protein complexes of bacterial effectors and host factors are scarce as training

material for AF2. In addition, even though the core structures of small G-proteins can be reliably modelled, AF2 fails in predicting the various conformational states of the switch regions due to an over-representation of the folded, GTP-state in the protein data bank. Thus, AF2 likely is unable to model different conformational states in particular for the complex interface. Since Rab1b's switch II region becomes profoundly restructured upon Lem3-binding, modelling of the interface fails. Consequently, our covalent capture approach is particularly superior to recent modelling approaches when transient complexes and complexes with structural rearrangements in the interface are considered.

Structure comparisons and functional experiments confirm that a conformational change in switch II, triggered by insertion of a hydrophobic β-hairpin into the hydrophobic core of Rab1b, is required for dephosphocholination. This finding provides an explanation for Lem3 showing a preference for the inactive, GDP-bound state. In addition, our complex structure explains why dephosphocholination of Rab1b is still possible for $Rab1b_{S76(PC)Y77(AMP)}$. $Y77_{Rab1b}$ is positioned in the hollow region of Lem3, which provides sufficient space for the AMP moiety.

There are functional similarities in the action of Lem3 in comparison to AnkX. Like $F215_{Lem3}$ on the β-hairpin replaces $Y78_{Rab1b}$ in the core of Rab1b during catalysis, AnkX also uses a hydrophobic thorn-like element to displace the same tyrosine, thereby unfolding and repositioning the switch II region. Furthermore, Lem3 and AnkX have evolved from the conserved activities of their respective enzyme families into new functions: The FIC-family enzyme AnkX catalyses phosphocholination, even though the FIC-family is mainly involved in protein AMPylation. Similarly, the PPM-family member Lem3 causes dephosphocholination, yet the PPM-family commonly acts as phosphatase. Another functional similarity between AnkX and Lem3 is the preference for the GDP-bound state of Rab1b, albeit less pronounced for AnkX[20].

In summary, covalent capture of the Lem3:Rab1b-complex in combination with structure determination revealed a local unfolding mechanism of Rab1b by Lem3, and provides insight into the target recognition of related PPM-phosphatases. This approach may serve as a template to characterising transient protein complexes in general.

## Methods

### Plasmid construction

AnkX constructs used in this publication were previously described in[20]. All Lem3 constructs were cloned into a modified pSF vector (Oxford Genetics) with an N-terminal $His_{10}$- and eGFP (green fluorescent protein)-Tag using SLIC (sequence and ligation independent cloning). Alike all Rab construct were cloned into a pMAL vector (New England Biolabs) with an N-terminal $His_6$- and MBP (maltose-binding protein)-Tag. Tags and POIs (protein of interest) were separated by a TEV (tobacco etch virus) protease cleavage site. Rab1b proteins intended for complex formation were additionally equipped with a $His_{10}$-Tag and a PreScission protease cleavage site resulting in the following order of tags from N- to C-Terminus: $His_6$-Tag, MBP-Tag, TEV protease cleavage site, 10x-His-Tag, PreScission protease cleavage site.

Point mutations were introduced using the Q5 Site-Directed Mutagenesis Kit (New England Biolabs) as described by the manufacturer.

### Protein expression and purification

All Lem3 and AnkX constructs were expressed in *Escherichia coli* BL21-CodonPlus (DE3)-RIL cells, Rab constructs were expressed in *E. coli* BL21-CodonPlus (DE3) cells[20]. Heat-shock transformation was used to transform the respective plasmid. A preculture in lysogeny broth (LB) medium with at least 10 colonies was grown for 4 h at 37 °C and 200 rpm (Infors HT shakers). 1 L cultures of LB with a starting $OD_{600}$ (optical density at 600 nm) of 0.04 were grown at 37 °C and 180 rpm until $OD_{600}$ 0.8. Protein expression was performed at 21 °C over night

and induced by adding 0.5 mM isopropyl-β-d-thiogalactopyranoside. The next day cells were harvested by centrifugation (5000 g, 30 min). Pellets were washed in ice-cold phosphate-buffered saline (PBS), centrifuged for 20 min at 3000 g and flash-frozen in liquid nitrogen. Pellets were stored at −80 °C.

For protein purification pellets were resuspended in Buffer $A_{Effector}$ (50 mM HEPES-NaOH pH 8, 500 mM NaCl, 1 mM $MgCl_2$, 5% Glycerol (v/v), 2 mM β-mercaptoethanol (β-ME)) for Lem3 and AnkX constructs or Buffer $A_{Rab}$ (50 mM HEPES-NaOH pH 7.5, 500 mM NaCl, 1 mM MgCl2, 10 µM GDP, 2 mM β-mercaptoethanol), mixed with a spatula tip of deoxyribonuclease I (DNAse I) (Sigma-Aldrich) and lysed using a French press system (Constant Cell Disruption Systems) at 1.8 kbar. Protein degradation by endogenous proteases was prohibited by adding 1 mM phenylmethylsulfonyl fluoride (PMSF). Cell lysates were cleared by centrifugation.

All chromatography steps during protein purification were performed using NGC medium-pressure liquid chromatography (LC) system (Bio-Rad Laboratories). Cleared lysates were supplemented with 25 mM imidazole and loaded on a 5 ml Nuvia IMAC column (Bio-Rad Laboratories). Proteins were washed using 30 mM (Rab proteins) or 40 mM (effector proteins) imidazole for 40 column volumes and eluted at 125–150 mM imidazole. Dialysis was performed over night at 4 °C against dialysis $buffer_{effector}$ (20 mM HEPES-NaOH pH 8, 50 mM NaCl, 1 mM MgCl2, 5% Glycerol (v/v), 2 mM β-mercaptoethanol (β-ME)) or dialysis $buffer_{Rab}$ (20 mM HEPES-NaOH pH 7.5, 50 mM NaCl, 1 mM MgCl2, 10 µM GDP, 2 mM β-mercaptoethanol (β-ME)) in presence of TEV protease. Removal of the solubility tags and His-Tagged TEV protease was achieved through either reverse metal chelate affinity chromatography (RMCAC) for effector proteins or RMCAC in combination with 5 ml MBPTrap HP column (GE Healthcare Life Sciences) for Rab proteins. SEC (size exclusion chromatography) was performed to separate oligomeric species in SEC $buffer_{effector}$ or SEC $buffer_{Rab}$ (20 mM HEPES-NaOH pH 8, 50 mM NaCl, 1 mM MgCl2, 5% Glycerol (v/v), 2 mM dithiothreitol (DTT)) or dialysis $buffer_{Rab}$ (20 mM HEPES-NaOH pH 7.5, 50 mM NaCl, 1 mM MgCl2, 10 µM GDP, 2 mM tris(2-carboxyethyl)phosphine (TCEP)). Proteins were concentrated to desired concentration using Amicon Ultra 15 ml centrifugal filters (Merck Millipore).

### Modification and purification of Rab1b and Rab35

Rab1b and Rab35 proteins were post-translationally modified using $AnkX_{1-800}$ as a transferase. For quantitative PCylation or PEylation of Rab proteins, 1 µM AnkX was incubated over night with 100 µM Rab protein and 250 µM CDP-Choline (Carbosynth) or CDP-ethanolamine (Jena Bioscience) respectively. For Phosphorylation (Pylation) 1 µM AnkX was incubated with 1 µM Rab and 100 µM CDP (Sigma) for 72 h. All modification reactions were performed at 19 °C in modification buffer [20 mM Hepes, 50 mM NaCl, 1 mM $MgCl_2$, 10 µM GDP, 1 mM TCEP (pH = 7.5)]. Rab modification using cysteine reactive CDP-Choline derivative was performed by incubating 1 µM AnkX, 500 µM Rab and 750 µM derivative. Modification took place for 24 h at 19 °C in adduct buffer [20 mM Hepes, 50 mM NaCl, 1 mM MgCl2, 10 µM GDP, 1 mM β-Me (pH = 7.5)]. The CDP-Choline derivative used in this work was previously described in[20].

AMPylation of $Rab1b_{3-174}$:GDP was performed as previously described in[13] with the following changes: AMPylation buffer (20 mM HEPES-NaOH pH 7.5, 50 mM NaCl, 1 mM MgCl2, 2 mM TCEP) was supplemented with 2.5 x M excess, compared to the Rab1b concentration, of GTP and 10 x M of ATP. $DrrA_{340-533}$ (GEF-domain) and $DrrA_{16-352}$ (ATase domain) were added in a 1:100 ratio to promote nucleotide exchange and AMPylation. Samples were incubated at 20 °C for 4 h. $Rab1b_{Y77(AMP)}$ was phosphocholinated as mentioned above.

Modified proteins were separated from AnkX/DrrA, free nucleotide and oligomers using size exclusion chromatography (SEC) (Cytiva

75 pg, 16/600). SEC was performed in the SEC buffer$_{Rab1b}$. Proteins were concentrated as described above, flash frozen in liquid nitrogen and stored at −80 °C.

## Analytical Lem3:Rab1b complex formation

All experiments regarding binary adduct formation and complex formation were performed in presence of β-ME as reducing conditions are required for activation of the cysteine preparing it for the reaction with the chloroacetamide. Other reducing reagents like DTT and TCEP were not used due to their, low abundant, reaction with the chloroacetamide group[20].

To investigate the capability of Lem3 and Lem3$_{Cys}$ to form the specific Lem3:Rab1b complex, equimolar amounts of Lem3 or Lem3$_{Cys}$ and Rab1b, modified with a cysteine reactive phosphocholine derivative, were incubated at 19 °C for 24 h.

Optimal conditions for complex formation was analysed in three steps. To analyse temperature dependence of complex formation, equimolar amounts of Lem3 and Rab1 were co-incubated at (19°, 25°, 30° or 37 °C) for 24 h. Complex formation over time was analysed using equimolar amounts of Lem3 and Rab1b which were incubated at 19 °C. Samples were taken after 1 min, 24 h, 48 h and 72 h. The effect of excess of Rab1 or Lem3 on complex yields was analysed by incubating equimolar amounts or 2 or 3 times excess of Rab1 or Lem3 at 19 °C for 24 h. All analytical complex formation experiments were performed in adduct buffer and complex formation was quantified by SDS-PAGE gel shift assay.

## Preparative Lem3:Rab1b complex formation

AnkX was used to quantitatively transfer PC-Cl to 10x-His-tagged-Rab1b from the cosubstrate CDP-Choline-Cl. Quantitative modification of Rab1b with the phosphocholine derivative was confirmed using intact MS. After modification with PC-Cl, the SEC-purified binary adduct Rab1b$_{S76(PC-Cl)}$ was co-incubated with equimolar amounts of Lem3$_{21-486}$ over night at 19 °C. Complex formation was confirmed using SDS-PAGE gel shift assay and 25 mM imidazole were added prior to loading on a 5 ml Nuvia IMAC column (Bio-Rad Laboratories). The column was washed with 50 mM imidazole for 400 ml to remove free Lem3. The complex and free Rab1b were eluted with 125 mM imidazole and dialysed against dialysis buffer$_{Rab1b}$ overnight in presence of Pre-Scission protease. SEC was performed to separate the complex from free Rab1b and oligomeric species. The complex was concentrated to desired concentration using Amicon Ultra 15 ml centrifugal filters (Merck Millipore).

## SDS-Page gel shift assay

To be analysed samples were boiled in 1x Laemmli buffer [50 mM tris, 10% (v/v) glycerol, 2% SDS, 200 mM β-ME, and 0.01% Bromophenol blue (pH 6.8)] for 5 min at 95 °C. Samples were separated in size on either 12% acrylamide gels or 4–15% gradient gels (Biorad). Gels were either stained using Roti-Blue quick (Roth) or with Coomassie Brilliant Blue.

## Analytical size exclusion chromatography

Quality analysis of purified complex was performed by analytical size exclusion. Protein samples (250 µg for Rab1b$_{S76T}$, 500 µg for Lem3$_{21-486}$ and Lem3:Rab1b complex) were injected to a preparative size exclusion chromatography column (16/600, 75 pg, Cytiva) and compared to a size exclusion chromatography standard (Bio-Rad Laboratories). All runs were performed on an Äkta prime system (GE Healthcare Life Sciences).

## Mass spectrometry-based activity assay of Lem3

Catalytic efficiencies (k$_{cat}$/K$_M$) for Lem3 were calculated based on assay derived demodification curves. Rab proteins (25 µM) were co-incubated with catalytic amounts of Lem3 (5 nM for Rab1b$_{S76(PC/PE)}$,

2.5 µM for Rab1b$_{S76(P)}$, 0.5 µM for Lem3$_{C134S\_C209S\_C395S\_C456S}$ and Lem3$_{C134S\_C209S\_T391C\_C395S\_C456S}$) in modification buffer. Samples were taken after 1 min, 5 min, 10 min, 30 min, 60 min, 120 min, 240 min, 360 min and 24 h and reaction was stopped by diluting the sample 1:5 in 1 mM EDTA. Samples were analysed via intact LC-MS (see MS analysis of purified proteins). Intensities for modified and unmodified protein were added and respective individual intensities were calculated in %. The resulting demodification curves were used to calculate the catalytic efficiencies. Observed rate constants were calculated using Eq. 1 and subsequently divided by the enzyme concentration to determine catalytic efficiencies (µM$^{-1}$ s$^{-1}$). All catalytic efficiencies are listed in the Source data. A comparable behaviour during LC-MS analysis for modified an unmodified Rab1b was proven by measuring an equimolar mixture and comparing resulting intensities. Correct mixture of this sample was monitored by analysing the individual proteins via SDS-PAGE (Supplementary Fig. 2b). Modified Rab1b behaves as the unmodified protein, excluding the possibility of biased data (Supplementary Fig. 2b, c).

$$y = y_0 + A \cdot e^{R_0 x} \tag{1}$$

Catalytic efficiency (k$_{cat}$/K$_M$) was calculated using Eq. 1, with y being the percentage of modified Rab1b, y$_0$ is the minimal modification rate, A is the amplitude, R$_0$ is the observed rate constant and x is time in [min]. Statistical significance was evaluated using the unpaired two-tailed Student's $t$ test. P values less than 0.05 were considered to be significant, $p$ values less than 0.01 were considered to be very significant and $p$ values less than 0.001 were considered to be highly significant.

## Mass spectrometric analysis of purified proteins

Purified protein samples (2 µl of 0.1 mg ml$^{-1}$ for Rab proteins, 2 µl of 0.3 mg ml$^{-1}$ for Lem3:Rab1b complex) were injected onto a ProSwift™ RP-4H 1 ×50 mm column (Thermo Fisher Scientific), coupled to an Elute UHPLC (Bruker). Desalted proteins were subsequently analysed by a maXis II ETD ESI LCMS (Bruker Daltonik). Data were evaluated using DataAnalysis (Version 5.1, Bruker Daltonics).

## Protein crystallisation and structure determination

**Lem3$_{21-486}$.** Crystals of Lem3$_{21-486}$ D190A were grown by sitting-drop vapor diffusion at 20 °C. Optimal crystals for structure determination were obtained from protein samples (0.2 µL) that were mixed with 0.2 µL of reservoir solution (0.1 M MES pH 6.5, 30% (v/v) PEG 300) and supplemented with 0.01 µL of a microseed solution (obtained from initial crystal screens) using an Oryx4 system (Douglas Instruments). Diffracting crystals grew within a few days and were cryoprotected by adding 1 µl of mother liquor containing 30% (v/v) glycerol before vitrification in liquid nitrogen.

To obtain suitable phases, crystals were soaked with trace amounts of thiomersal (Hg scattering) for two hours prior to cryoprotection. A data set of an Hg-soaked crystal was detected using synchrotron radiation (λ = 1.0 Å, f´ = −19.4, f´´ = 10.2) at Beamline X06SA (Swiss Light Source, Paul Scherrer Institute, Villigen, Switzerland). The diffraction intensity data were analysed using XDS (Version 2010)[44] and three Hg-sites were determined by SHELX (Version 2001)[45]. Phases were calculated with SHARP (Version 2.0)[46] and improved with the Phenix (Version 1.14) software package[47]. An initial model was built with COOT (Version 0.8.6)[48].

This low-resolution structure was used for molecular replacement by Patterson search calculations applying Phaser (Version 3.0)[49] and using diffraction intensity data from a native crystal (Supplementary Table 1). Resolution limits were chosen to meet the following criteria: I/σ(I) > 2.0, R$_{merge}$ < 70%, and redundancy >3.0. The structure model was optimised in COOT (Version 0. 8.6) with intermittent constrained refinements using REFMAC5 (Version 5.7)[50]. Water molecules were

positioned using ARP/wARP solvent (Version 7)[51]. TLS (translation/libration/screw) and restricted refinements yielded appropriate $R_{work}$ and $R_{free}$ ratios and root-mean-square deviation (RMSD) values for bond lengths and angles (validated by PROCHECK[52]). Data collection and refinement statistics of this and the other crystal structures presented here are gathered in Supplementary Table 1.

**Lem3$_{FL}$.** Hanging drop crystallisation trials were carried out at 19 °C, by mixing equal volumes (1 μL) of reservoir solution and protein solution. Crystals grew in a condition containing MES 0.1 M pH 5, PEG6000 5%. Crystals were soaked in cryo-solutions containing the crystallisation mother liquor supplemented with 25% [v/v] glycerol, mounted onto a cryoloop (Hampton Research), and immediately flash-cooled in liquid nitrogen. Diffraction data were collected at EMBL beamline P13 at the PETRA III storage ring (DESY, Hamburg, Germany). Diffraction data were processed using XDS (Version 2021)[44] and scaled with Aimless (Version 0.7.4) from the CCP4 suite (Version 7.1.)[53,54].

The structures were solved by molecular replacement with Phaser (Version 2.8)[49] using Lem3$_{21-486}$ as a search model. The obtained model solution was then extended with the help of the Lem3$_{FL}$ model from AlphaFold2 and further corrected manually with COOT (Version 0.9.8)[48] and refined using the PHENIX suite (Version 1.19)[47]. The quality of the final model was assessed using the wwPDB validation server[55] and Molprobity[56].

**Lem3:Rab1b complex.** Sitting drop crystallisation trials were carried out at 19 °C, by mixing equal volumes (100 nl) of reservoir solution and protein solution. Crystals grew in a condition containing 40% (v/v) PEG600, 0.2 M calcium acetate, 0.1 M sodium cacodylate pH 6.5. Crystals were soaked in cryo-solutions containing the crystallisation mother liquor supplemented with 25% [v/v] glycerol, mounted onto a cryoloop (Hampton Research), and immediately flash-cooled in liquid nitrogen. Diffraction data were collected at EMBL beamline P13 at the PETRA III storage ring (DESY, Hamburg, Germany). Diffraction data were processed using XDS (Version 2021)[44] and scaled with Aimless (Version 0.7.4) from the CCP4 suite (Version 7.1)[53,54].

The complex structure was solved by molecular replacement with Phaser (Version 2.8)[49] using search models based on structures Rab1 (PDB ID: 3NKV) and Lem3$_{21-486}$. The obtained model solution was then corrected and further built manually with COOT (Version 0.9.8)[48] and refined using the PHENIX suite (Version 1.19)[47] and the PDB_REDO web server[57]. The quality of the final model was assessed using the wwPDB validation server[55] and Molprobity[56].

### Nano differential scanning fluorimetry (nanoDSF)
All Lem3 and Rab1 variants were diluted to 0.2 mg ml$^{-1}$ in respective SEC$_{buffer}$ complemented or not with 1 mM metal ions (Mg$^{2+}$, Mn$^{2+}$, or Ca$^{2+}$). The samples were loaded into the standard capillaries (Nanotemper, #PR-C002) and analysed via the Prometheus NT.48 (Nanotemper) (Software: PR.ThermControl) at a gradient of 1 °C per min ranging from 15 to 85 °C. The melting points were derived from the ratio of fluorescence at 350/330 nm.

All mutants were analysed regarding their thermal stability to exclude the possibility of reduced catalytic efficiency rates due to unstable or misfolded protein (Supplementary Fig. 6e, f).

### Phos-Tag gel electrophoresis
Phos-Tag gel electrophoresis was performed as described in ref. 58.

### Structural alignment of PPM phosphatases
The structural comparison of PPM phosphatases for the generation of Supplementary Fig. 5c was done in PyMOL (Version: PyMOL(TM) 2.3.2, https://pymol.org/2/).

### Reporting summary
Further information on research design is available in the Nature Portfolio Reporting Summary linked to this article.

## Data availability
Structure factors and model coordinates for Lem3 have been deposited in the Protein Data Bank (PDB) under the accession code 8ANP and 8AGG, for the Lem3:Rab1b complex under the accession code 8ALK. All other data related to this paper are available from the corresponding author. Source data are provided with this paper. Crystal structure data used in this study and published elsewhere are also available in the PDB, deposited with the following accession codes: 4RA2, 6RRE, 6B67, and 3NKV. Source data are provided with this paper.

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

## Acknowledgements

Mass spectrometry was funded by the Deutsche Forschungsgemeinschaft (DFG, German Research Foundation – Projektnummer INST 152/859-1 FUGG, A.I.). This work was performed within the framework of SFB 1035 (German Research Foundation DFG, Sonderforschungsbereich 1035, Projektnummer 201302640, projects A02, M.G., and B05, A.I.). C.H. thanks Knut and Alice Wallenberg Foundation Sweden (KAW 2013.0187) and Swedish Research Council (VR) for generous support. The synchrotron MX data were collected at beamlines P13 operated by EMBL Hamburg at the PETRA III storage ring (DESY, Hamburg, Germany) and X06SA at the Swiss Light Source (Villigen, Switzerland). We would like to thank the local contacts for the assistance in using the beamlines. We are grateful to the Swedish Research Council (VR grant: 2019-05384, C.H.) for generous support. We also acknowledge technical support from the SPC facility at EMBL Hamburg. A.I. acknowledges access to the core facilities and laboratories of the Centre

for Structural Systems Biology (CSSB, Hamburg). We acknowledge financial support from the Open Access Publication Fund of UKE - Universitätsklinikum Hamburg-Eppendorf- and DFG – German Research Foundation. We thank Dr. Dorothea Höpfner for advice during development of the MS-based activity assay for Lem3. We acknowledge help of Daniel Otero with protein expression.

## Author contributions

S.E. and F.E. obtained the crystal structures of Lem321-486. M.G. and F.E. solved and refined the Lem321-486 crystal structure (PDB ID: 8ANP). M.S.K., A.I and C.H. envisioned the strategy to obtain the Lem3:Rab1b complex. M.S.K. and V.P. obtained and refined the crystal structure of Lem3FL and the Lem3:Rab1b complex. M.S.K., A.I. and V.P. analysed and interpreted all protein structures. M.S.K. purified the Lem3:Rab1b complex, developed and performed all biochemical experiments presented in this paper, except nanoDSF measurements which were performed by V.P. C.H., C.P. and P.O. provided the CDP-Choline-Cl. M.S.K. and V.P. prepared all figures in this paper. M.S.K., V.P. and A.I. wrote the paper. A.I., M.G. and C.H. provided laboratory infrastructure. All authors participated in manuscript editing and final approval.

## Funding

## Competing interests

The authors declare no competing interests.
