## [Peer Review File · Nature Communications]

REVIEWER COMMENTS

Reviewer #1 (Remarks to the Author):

The manuscript by Kaspers et al. reported the crystal structures of a Legionella effector Lem3, which specifically removes the phosphocholine group from Rab1 as a dephosphocholine enzyme. The authors also stabilized the transient Lem3:Rab1b complex using a covalent crosslinker and determined the structure of this cross-linked complex. Structural analysis, combined with mutagenesis and enzymatic assays revealed the catalytic mechanism of dephosphocholine mediated by Lem3. This mechanism provides insight into the recognition of substrates by other PPM phosphatases. Overall, this paper presented a cohort of solid structural and biochemical data. However, the scope of the paper is narrow. Lem3 is the only enzyme that catalyzes dephosphocholine. It has limited broad interest to general readers.

Minor

1. phosphoethanolamine usually takes the acronym PE instead of PA, which is phosphatidic acid.

Reviewer #2 (Remarks to the Author):

This study from Dr. Itzen's group provides a biochemical and structural characterization of the Legionella effector protein Lem3, which has dephosphocholinase activity. During infection, Legionella post-translationally modify the small GTPase Rab1 to achieve its hyperactivation and hijack intracellular trafficking of the host cell. Phosphocholine is one of these PTMs, which is later reverted by Lem3.

To gain insight into Lem3 function, the authors determine its structure by X-ray crystallography in the apo form and bound to phosphocholinated Rab1. Here, the enzyme-substrate complex is captured by an elegant covalent coupling approach. This required the modification of phosphocholine with chloroacetamide as well as introduction of in total five mutations in Rab1b and Lem3. The structural work is accompanied by enzymatic studies with different substrates of WT Lem3 as well as mutants that were designed to verify the proposed mechanism.

Overall, the work presented here provides interesting insight into the catalytic mechanism of this unique enzyme, but also the molecular basis of substrate specificity of PPM phosphatases. However, several points should be addressed before publication:

1. There is one major issue that needs to be clarified in a revised manuscript: the data presented in Fig4D shows no activity of Lem3 towards Rab1 and Rab35 when the cholinated residue is a Thr, but same activity when the cholinated residue is Ser. On the other hand, the inactive construct Rab1b76T(PC) was used for the covalent capture and formed the supposedly active complex. This is a discrepancy that indicates a) an error in the enzymatic assays, or b) that the Rab1b(PC-Cl)-Lem3 structure dose not represent an active state.

To resolve this, activity assays with Rab1b76S(PC) and Rab1b76T(PC) should be repeated with Lem3 WT and the crystallized variant Lem3 C134S/C209S/T391C/C456S as well as Lem3 C134S/C209S/C456S.

It would also be important to compare the Lem3-interacting residues of Rab1b with the corresponding switch II residues of Rab35: is binding of Rab35 to Lem3 possible?

Additional points:

2. There is some redundancy in the structure panels, e.g. Fig 1D and S1B are basically identical and could be both omitted. Fig 1B, 1C and S1A could also be combined. Fig S1C(left) and 2B also provide the same information.

Along these lines, the description of the Lem3 and Rab1(PC-Cl):Lem3 structures (p. 3-8 and p 17-19) could be streamlined to appeal to a broader readership.

3. It would be informative to include key interacting residues in Fig 6C. For the discussion of the putative mechanism (lines 383-388) a schematic representation may also be helpful.

4. Most activity assays (Fig 3B, 4B, 4D, 7A-F) are normalized to catalytic efficiency (%). The absolute values should be shown.

Minor issues:

- Fig 2A: the PDB ID for Lem3 21-486 is incorrect

- Table S1: please provide multiplicity and CC1/2 for 8ANP for total data and outer shell

- Lines 245 and 248: Rab1b S76T(PC) should be Rab1b S76T(PC-Cl)

- Line 338: Y372ALem3 (12.4 %) does not fit the data in Fig 7B

- Fig S5E,F: if these data are technical replicates with the same protein preparation, error bars are not appropriate

- Line 575: the equation allows determination of observed rate constants. The calculation of catalytic efficiency (kcat/KM) is missing in the method section

Response to referees

Dephosphocholination by Legionella effector Lem3 functions through remodelling of the switch II region of Rab1b

We would like to thank the referees for their time and effort. We appreciate their constructive comments on our manuscript. Based on the referee comments we have optimized several aspects of our findings and have particularly improved the presentation of the data and the clarity of our arguments. We hope that the referees will concur with the changes to the manuscript.

Reviewer #1
The manuscript by Kaspers et al. reported the crystal structures of a Legionella effector Lem3, which specifically removes the phosphocholine group from Rab1 as a dephosphocholination enzyme. The authors also stabilized the transient Lem3:Rab1b complex using a covalent crosslinker and determined the structure of this cross-linked complex. Structural analysis, combined with mutagenesis and enzymatic assays revealed the catalytic mechanism of dephosphocholination mediated by Lem3. This mechanism provides insight into the recognition of substrates by other PPM phosphatases. Overall, this paper presented a cohort of solid structural and biochemical data. However, the scope of the paper is narrow. Lem3 is the only enzyme that catalyses dephosphocholination. It has limited broad interest to general readers.
We thank reviewer #1 for the positive reception of our manuscript. We would like to emphasize that the structure Lem3 is homologous to structures of PPM phosphatases. Therefore, even though Lem3 is a dephosphocholinase, the basic mechanism of action is likely similar to the PPM-family of phosphatases. Therefore, we believe that our data will also stimulate research into the action of these enzymes. Also, the Lem3:Rab1-complex structure is the first example of a full protein substrate with its corresponding PPM-type phosphatase/dephosphocholinase and thus is also of general interest.
Minor
1. phosphoethanolamine usually takes the acronym PE instead of PA, which is phosphatidic acid.
As requested, we have modified the abbreviation for phosphoethanolamine in the manuscript.

Reviewer #2

This study from Dr. Itzen's group provides a biochemical and structural characterization of the Legionella effector protein Lem3, which has dephosphocholinase activity. During infection, Legionella post-translationally modify the small GTPase Rab1 to achieve its hyperactivation and hijack intracellular trafficking of the host cell. Phosphocholination is one of these PTMs, which is later reverted by Lem3.

To gain insight into Lem3 function, the authors determine its structure by X-ray crystallography in the apo form and bound to phosphocholinated Rab1. Here, the enzyme-substrate complex is captured by an elegant covalent coupling approach. This required the modification of phosphocholine with chloroacetamide as well as introduction of in total five mutations in Rab1b and Lem3. The structural work is accompanied by enzymatic studies with different substrates of WT Lem3 as well as mutants that were designed to verify the proposed mechanism. Overall, the work presented here provides interesting insight into the catalytic mechanism of this unique enzyme, but also the molecular basis of substrate specificity of PPM phosphatases. However, several points should be addressed before publication:

We are grateful for the positive feedback by reviewer #2 and we appreciate his/her recognition of the importance of our work on the catalytic mechanism and substrate specificity of PPM phosphatases. We have carefully considered the issues raised by reviewer #2 and have included additional experimental data and clarifications in our revised manuscript that address these concerns in full.

1. There is one major issue that needs to be clarified in a revised manuscript: the data presented in Fig4D shows no activity of Lem3 towards Rab1 and Rab35 when the cholinated residue is a Thr, but same activity when the cholinated residue is Ser. On the other hand, the inactive construct Rab1b76T(PC) was used for the covalent capture and formed the supposedly active complex. This is a discrepancy that indicates a) an error in the enzymatic assays, or b) that the Rab1b(PC-Cl)-Lem3 structure does not represent an active state.

To resolve this, activity assays with Rab1b76S(PC) and Rab1b76T(PC) should be repeated with Lem3 WT and the crystallized variant Lem3 C134S/C209S/T391C/C456S as well as Lem3 C134S/C209S/C456S.

It would also be important to compare the Lem3-interacting residues of Rab1b with the corresponding switch II residues of Rab35: is binding of Rab35 to Lem3 possible?

Indeed, the complex is not capable of cleaving the phosphocholine from Rab1b_{76T}. Our reasoning for referring to the complex as active is because Lem3 in the complex is an active enzyme and no mutation is introduced in it to interfere with the enzymatic activity. Rather, we have exploited the peculiar fact that Lem3 is unable to remove the phosphocholine group when it is attached to the threonine in Rab1. Therefore, we consider the complex active, but hydrolysis-deficient and preserving the details on protein-protein interaction between the Lem3 and Rab1b.

Realising that our statement may cause confusion, we address the reviewers concern via inclusion of a clarification. We have changed the text hence as follows:

"Since the mutation D190A_{Lem3} was not sufficient to prevent cleavage of the phosphodiester, we use the S76T-substitution in Rab1b (Rab1b_{S76T(PC-Cl)}) for further complex formation. This allowed us to exploit the inability of Lem3 to cleave the phosphate at threonine residues, thereby gaining a hydrolysis-deficient complex while maintaining Lem3 as an active enzyme."

Additionally, we incorporated the following statements into the discussion:

"Although we crystallised a hydrolysis deficient complex due to the S76T mutation in Rab1b, the Lem3:Rab1b crystal structure can be used to analyse the active centre since it contains the active Lem3 variant." "Although the hydrolysis-deficient complex structure cannot represent the active enzymatic state due to the inability of Lem3 to cleave threonine-PC-bonds, our data suggest that the complex structure represents a snapshot of the actual complex binding event of Rab1b by Lem3."

Regarding the second point, we have carefully considered the suggestion to analyse the catalytic activities of specific Lem3 mutants and have chosen to study the Lem3_{C134S_C209S_T391C_C395S_C456S} and Lem3_{C134S_C209S_C395S_C456S} mutants because they are used in the crystallisation and control construct, respectively. We have included data on the above-mentioned mutants in the revised manuscript (Figure S4c-e).

Furthermore, we have also supplemented the manuscript with a new figure (Fig. S3) demonstrating the conservation of sequence and structure between Rab1b and Rab35. Please note that data on demodification of the two wild type proteins and the Rab1b_{S76T}/Rab35_{T76S} mutants by Lem3 were already included in the manuscript (Fig. 4d).

In addition, as requested, we have also included an experiment demonstrating that complex formation between Lem3 and Rab35 is possible using our approach. The yields of covalent complex formation are comparable to those observed between Rab1b_{S76T} and Lem3 (Fig. S3c).

Additional points:

2. There is some redundancy in the structure panels, e.g., Fig 1D and S1B are basically identical and could be both omitted. Fig 1B, 1C and S1A could also be combined. Fig S1C(left) and 2B also provide the same information.

Along these lines, the description of the Lem3 and Rab1(PC-Cl):Lem3 structures (p. 3-8 and p 17-19) could be streamlined to appeal to a broader readership.

We have followed the Reviewers' suggestions and replaced Fig. 1d with Fig. S1b, as Fig. S1b offers more information. In addition, we have combined Fig. 1b and 1c and removed Fig. S1c (left). We did not include Fig. S1a in the newly combined figure as it would have been difficult to assign the individual structural elements clearly.

Regarding the description of the Lem3 structure, we have condensed the section "The Lem3 crystal structure" as follows:

"In order to obtain structural insights into Lem3, we crystallised the full-length protein comprising the amino acids (aa) 1-570 (Lem3FL) and a shortened construct (aa 21-486, Lem321-486). Lem321-486 possesses full catalytic activity in regard to dephosphocholination of Rab1b phosphocholinated at Ser76, indicating that the N- and C-terminal regions of Lem3 are not involved in catalysis (Fig. 1a). We solved the structures of Lem3FL and Lem321-486 at 3.6 Å and 2.2 Å resolution, respectively. The Lem321-486 structure was solved experimentally using the anomalous dispersion from heavy atoms incorporated to the protein by soaking (Table S1). In parallel, Lem321-486 was modelled using AlphaFold2 (AF2) 26. A superimposition of the Lem321-486 crystal and the AF2-predicted structure revealed high similarity (0.88 Å RMSD (root-mean-square deviation) on 444 superimposed Ca-atoms) (Fig. S1a). Furthermore, we were able to solve the structure of Lem3FL by molecular replacement using the short construct structure and further extend the building of the C-terminal part with help of the AF2 model of Lem3FL (Fig. 1b, Table S1) 27.

Overall, Lem3 has a scalene triangular shape, formed by 17 β-strands and 20 α-helices (Fig. 1b-c). Metaphorically, the core domain folds into a fist and a raised thumb, the latter consisting of helices α9-α13. The extended α-helix α9 appears to play the role of the first metacarpal bone which links the thumb to the palm. Lem3FL extends the fist opposite from the thumb by three α-helices (α18-

$\alpha 20$, aa 495-567). Together with α -helices $\alpha 14$ - $\alpha 17$, these three helices form a helical bundle consisting of helices $\alpha 14$ - $\alpha 20$ (Fig. 1b-c).”

However, we decided to keep the complex structure description in full since we want to make this topic comprehensible also for readers that are interested in the mechanism of PPM-type phosphatases in general.

3. It would be informative to include key interacting residues in Fig 6C. For the discussion of the putative mechanism (lines 383-388) a schematic representation may also be helpful.

We intended Fig. 6c to demonstrate the linker and the linked amino acids, as well as the distance between the linked amino acids rather than individual interactions. As suggested by the reviewer, we now include the depiction of L68_{Lem3}, the only amino acid interacting with the linker apart from T391_{Lem3}. The latter is mutated to cysteine to react with the chloroacetamide function of the linker; it has therefore been already displayed in the depiction.

A schematic representation of the proposed mechanism has been added to the manuscript as suggested by the Reviewer in Fig. S5f.

4. Most activity assays (Fig 3B, 4B, 4D, 7A-F) are normalized to catalytic efficiency (%). The absolute values should be shown.

We chose to normalise all catalytic efficiencies to the wild type catalytic efficiency (set at 100%) in order to make direct comparison of the different mutants more accessible. While we prefer this representation, we understand the reasoning of providing the absolute values. We have therefore included an extra folder within the “Source data” file displaying the conversion rates and the calculations of % values.

Figure	Panel description	Additional information	Control efficiency (%)	Control	Control	Control	Control	Control
Fig 2a	Lem3 ^{WT} Lem3 ^{Y372A}		0.0457 0.0516 0.0703 0.056	92	94	93	94	93
Fig 2b	Lem3 ^{WT} Lem3 ^{WT}	Magnesium Phosphorus Calcium	0.0517 0.0528 0.0621 0.056 0.126 0.166 0.192 0.121 n.d. n.d. n.d. n.d.	96	98	97	98	97
Fig 4b	Lem3 ^{WT} Lem3 ^{WT} Lem3 ^{WT}	Rab35 mutants Rab35 mutants Rab35 mutants	0.0487 0.0498 0.0503 0.056 0.0290 0.0310 0.0480 0.036 0.0206 0.0096 0.0095 0.019	92	93	92	93	92
Fig 4d	Lem3 ^{WT} Lem3 ^{WT} Lem3 ^{WT}	Rab35 mutants Rab35 mutants Rab35 mutants	n.d. n.d. n.d. n.d. 0.0510 0.0520 0.0509 0.052 0.0447 0.0393 0.0444 0.042	n.d.	n.d.	n.d.	n.d.	n.d.
Supplemental Fig 4d	Lem3 ^{WT} Lem3 ^{WT} Lem3 ^{WT}	Rab35 mutants Rab35 mutants Rab35 mutants	0.0759 0.0889 0.0746 0.080 0.0006 0.0007 0.0007 0.001 0.0010 0.0003 0.0003 0.000	95	92	93	95	94
Fig 7a	Lem3 ^{WT} Lem3 ^{WT}	Rab1b GDP Rab1b GDP	0.0738 0.0740 0.0758 0.074 0.0249 0.0246 0.0242 0.023	98	90	92	98	92
Fig 7b	Lem3 ^{WT} Lem3 ^{WT}	Rab1b GDP Rab1b GDP	0.0209 0.0242 0.0262 0.024 0.0635 0.0630 0.0679 0.062	92	93	94	92	94
Fig 7c	Lem3 ^{WT} Lem3 ^{WT}	Rab1b mutants Rab1b mutants Rab1b mutants	0.0090 0.0096 0.0091 0.0091 0.0340 0.0306 0.0363 0.036 0.0001 0.0001 0.0001 0.000	96	98	94	96	94
Fig 7d	Lem3 ^{WT} Lem3 ^{WT}	Rab1b mutants Rab1b mutants Rab1b mutants	0.0050 0.0051 0.0051 0.005 n.d. n.d. n.d. n.d. 0.0245 0.0234 0.0261 0.025	9	9	9	9	9
Fig 7e	Lem3 ^{WT} Lem3 ^{WT}	Rab1b mutants Rab1b mutants Rab1b mutants	0.0092 0.0092 0.0090 0.009 0.0240 0.0262 0.0230 0.024 0.0236 0.0276 0.0264 0.024	95	95	96	95	96
Fig 7f	Lem3 ^{WT} Lem3 ^{WT}	Rab1b mutants Rab1b mutants Rab1b mutants	0.0008 0.0007 0.0070 0.002 0.0012 0.0006 0.0024 0.006 0.0008 0.0002 0.0021 0.001	33	53	50	33	50
Fig 7g	Lem3 ^{WT} Lem3 ^{WT}	Rab1b mutants Rab1b mutants Rab1b mutants	0.0017 0.0017 0.0017 0.001 0.0017 0.0017 0.0017 0.001 0.0017 0.0017 0.0017 0.001	6	7	7	6	7
Fig 7h	Lem3 ^{WT} Lem3 ^{WT}	Rab1b mutants Rab1b mutants Rab1b mutants	0.0017 0.0017 0.0017 0.001 0.0017 0.0017 0.0017 0.001 0.0017 0.0017 0.0017 0.001	92	90	90	92	90
Fig 7i	Lem3 ^{WT} Lem3 ^{WT}	Rab1b mutants Rab1b mutants Rab1b mutants	0.0017 0.0017 0.0017 0.001 0.0017 0.0017 0.0017 0.001 0.0017 0.0017 0.0017 0.001	5	2	2	5	2
Fig 7j	Lem3 ^{WT} Lem3 ^{WT}	Rab1b mutants Rab1b mutants Rab1b mutants	0.0017 0.0017 0.0017 0.001 0.0017 0.0017 0.0017 0.001 0.0017 0.0017 0.0017 0.001	30	43	43	30	43
Fig 7k	Lem3 ^{WT} Lem3 ^{WT}	Rab1b mutants Rab1b mutants Rab1b mutants	0.0017 0.0017 0.0017 0.001 0.0017 0.0017 0.0017 0.001 0.0017 0.0017 0.0017 0.001	59	78	79	59	79
Fig 7l	Lem3 ^{WT} Lem3 ^{WT}	Rab1b mutants Rab1b mutants Rab1b mutants	0.0017 0.0017 0.0017 0.001 0.0017 0.0017 0.0017 0.001 0.0017 0.0017 0.0017 0.001	91	92	92	91	92
Fig 7m	Lem3 ^{WT} Lem3 ^{WT}	Rab1b mutants Rab1b mutants Rab1b mutants	0.0017 0.0017 0.0017 0.001 0.0017 0.0017 0.0017 0.001 0.0017 0.0017 0.0017 0.001	7	9	9	7	9
Fig 7n	Lem3 ^{WT} Lem3 ^{WT}	Rab1b mutants Rab1b mutants Rab1b mutants	0.0017 0.0017 0.0017 0.001 0.0017 0.0017 0.0017 0.001 0.0017 0.0017 0.0017 0.001	9	9	9	9	9
Fig 7o	Lem3 ^{WT} Lem3 ^{WT}	Rab1b mutants Rab1b mutants Rab1b mutants	0.0017 0.0017 0.0017 0.001 0.0017 0.0017 0.0017 0.001 0.0017 0.0017 0.0017 0.001	96	97	96	96	96
Fig 7p	Lem3 ^{WT} Lem3 ^{WT}	Rab1b mutants Rab1b mutants Rab1b mutants	0.0017 0.0017 0.0017 0.001 0.0017 0.0017 0.0017 0.001 0.0017 0.0017 0.0017 0.001	92	92	92	92	92
Fig 7q	Lem3 ^{WT} Lem3 ^{WT}	Rab1b mutants Rab1b mutants Rab1b mutants	0.0017 0.0017 0.0017 0.001 0.0017 0.0017 0.0017 0.001 0.0017 0.0017 0.0017 0.001	52	21	20	52	20
Fig 7r	Lem3 ^{WT} Lem3 ^{WT}	Rab1b mutants Rab1b mutants Rab1b mutants	0.0017 0.0017 0.0017 0.001 0.0017 0.0017 0.0017 0.001 0.0017 0.0017 0.0017 0.001	93	85	86	93	85
Fig 7s	Lem3 ^{WT} Lem3 ^{WT}	Rab1b mutants Rab1b mutants Rab1b mutants	0.0017 0.0017 0.0017 0.001 0.0017 0.0017 0.0017 0.001 0.0017 0.0017 0.0017 0.001	90	83	83	90	83

In the spirit of transparency, we would like to add that we spotted an error in the calculation of one value related to Fig. 4d: The catalytic efficiencies of Lem3 for Rab35^{T76S(PC)} had accidentally been normalised towards the mean of the data corresponding to Rab35^{T76S(PC)} instead towards the control Rab1b^{S76(PC)}. This mistake does not affect the outcomes and the conclusions drawn from this experiment since Lem3 is (as stated previously) able to demodify Rab35^{T76S(PC)} with comparable rates. We apologise for this oversight.

Minor issues:

- Fig 2A: the PDB ID for Lem3 21-486 is incorrect

We are grateful to the reviewer for their careful reading of the manuscript. We have now corrected this error.

- Table S1: please provide multiplicity and CC1/2 for 8ANP for total data and outer shell

The missing values are now included in Table S1.

- Lines 245 and 248: Rab1b S76T(PC) should be Rab1b S76T(PC-Cl)

We agree that none of the terms used previously accurately describe the corresponding cross-linked product. The product consists of Rab1b linked to phosphocholine fused to a C3 linker, which - after reaction - links with the Lem3-cysteine. Rab1b^{S76T(PC)} refers to Rab1b^{S76T} phosphocholinated at Thr76, while Rab1b^{S76T(PC-Cl)} refers to Rab1b^{S76T} modified with the phosphocholine-chloroacetamide group. After reacting with a cysteine, the chlorine is eliminated from the resulting product. Therefore, we propose the new term Rab1b^{S76T(PC-C3)} to describe this modification more accurately. We appreciate the reviewer's suggestion to clarify this issue.

- Line 338: Y372ALem3 (12.4 %) does not fit the data in Fig 7B

As shown in the "Source data", Lem3^{Y372} has an activity of 0.124%. We have corrected this typo.

- Fig S5E,F: if these data are technical replicates with the same protein preparation, error bars are not appropriate

As suggested by the reviewer, we have removed the error bars from Fig. S5e and f (now Fig. S6e-f).

- - Line 575: the equation allows determination of observed rate constants. The calculation of catalytic efficiency (k_{cat}/K_M) is missing in the method section

The method section has been updated as suggested and now includes a sentence describing the calculation of catalytic efficiencies using observed rate constants.

REVIEWERS' COMMENTS

Reviewer #2 (Remarks to the Author):

The authors have carefully addressed my comments and questions and provided helpful additional data.

One point remains to be clarified: new Figure S4d shows that the Lem3 mutant used for crystallization has <1% activity at 100x protein concentration compared to wild-type and thus does not qualify as an active enzyme. I therefore recommend removing the statement "while maintaining Lem3 as an active enzyme" (l. 218).

The loss of activity of the crystallized Lem3 with all catalytic residues present, and potentially also the inability of WT to demodify Rab1b (S76T-PC), indicates some distortion of substrate binding to the active site as an effect of the mutations. The statement "Thus, the mutations do not affect the overall structure of Lem3." (l. 229) should be extended by "although some distortions in the active site may occur that cause reduction of activity" or something alike.

This does not question the rationale of the covalent capture approach or the conclusions presented in the paper and I therefore recommend acceptance of the manuscript after minor changes to the text.

Response #2 to referees

Dephosphocholination by Legionella effector Lem3 functions through remodelling of the switch II region of Rab1b

Reviewer #2
The authors have carefully addressed my comments and questions and provided helpful additional data.
One point remains to be clarified: new Figure S4d shows that the Lem3 mutant used for crystallization has <1% activity at 100x protein concentration compared to wild-type and thus does not qualify as an active enzyme. I therefore recommend removing the statement "while maintaining Lem3 as an active enzyme" (l. 218).
We removed the statement "while maintaining Lem3 as an active enzyme" as suggested.
The loss of activity of the crystallized Lem3 with all catalytic residues present, and potentially also the inability of WT to demodify Rab1b (S76T-PC), indicates some distortion of substrate binding to the active site as an effect of the mutations. The statement "Thus, the mutations do not affect the overall structure of Lem3." (l. 229) should be extended by "although some distortions in the active site may occur that cause reduction of activity" or something alike. This does not question the rationale of the covalent capture approach or the conclusions presented in the paper and I therefore recommend acceptance of the manuscript after minor changes to the text.
We have adjusted the statement so that it now reads as follows: " Thus, the mutations do not affect the overall structure of Lem3 or the arrangement of amino acids within the active site, although minor distortions may occur that cause reduction of activity. "